# Lack of the MHC class II chaperone H2-O causes susceptibility to autoimmune diseases

**Robin A. Welsh**[1☉], **Nianbin Song**[1☉], **Catherine A. Foss**[2], **Tatiana Boronina**[3], **Robert N. Cole**[3], **Scheherazade Sadegh-Nasseri**[1,4]*

**1** Graduate Program in Immunology, Johns Hopkins University School of Medicine, Baltimore, Maryland, United States of America, **2** Russel H. Morgan Department of Radiology and Radiological Science, Johns Hopkins University School of Medicine, Baltimore, Maryland, United States of America, **3** Mass Spectrometry and Proteomics Core, Johns Hopkins University School of Medicine, Baltimore, Maryland, United States of America, **4** Department of Pathology, Johns Hopkins University School of Medicine, Baltimore, Maryland, United States of America

☉ These authors contributed equally to this work.
* ssadegh@jhmi.edu

## Abstract

DO (HLA-DO, in human; murine H2-O) is a highly conserved nonclassical major histocompatibility complex class II (MHC II) accessory molecule mainly expressed in the thymic medulla and B cells. Previous reports have suggested possible links between DO and autoimmunity, Hepatitis C (HCV) infection, and cancer, but the mechanism of how DO contributes to these diseases remains unclear. Here, using a combination of various in vivo approaches, including peptide elution, mixed lymphocyte reaction, T-cell receptor (TCR) deep sequencing, tetramer-guided naïve CD4 T-cell precursor enumeration, and whole-body imaging, we report that DO affects the repertoire of presented self-peptides by B cells and thymic epithelium. DO induces differential effects on epitope presentation and thymic selection, thereby altering CD4 T-cell precursor frequencies. Our findings were validated in two autoimmune disease models by demonstrating that lack of DO increases autoreactivity and susceptibility to autoimmune disease development.

**Data Availability Statement:** All relevant data are within the paper and its Supporting Information files.

## Introduction

Autoimmune diseases are complex and multifaceted disorders. While extensive research has gone into understanding the underlying causes of various autoimmune pathologies, one of the first risk factors described was the Major Histocompatibility Complex (MHC) loci in disease development [1,2]. For example, a main risk factor to developing such autoimmune conditions as: rheumatoid arthritis (RA), type 1 diabetes (T1D), and celiac disease (CD) are specific MHC class II alleles (RA/T1D: HLA-DR1, CD: HLA-DQ2.5/8) [3]. However, much remains unknown as to why certain MHC II alleles associate with these autoimmune diseases. To explore the answer to this question, major research efforts have been invested into understanding the MHC class II processing pathway. Among all the components in the MHC class II processing pathways, two highly conserved nonclassical class II accessory molecules, DM (HLA-DM human; murine H2-M) and DO, are directly involved in the regulation and editing

**Funding:** This work was funded by grants from the National Institutes of Allergy and Infection Diseases: R21AI101987, R01AI120634 (SS-N), and AAI Careers in Immunology Fellowship Award (SS-N). The funders had no role in study design, data collection and analysis, decision to publish, or preparation of the manuscript.

**Competing interests:** The authors have declared that no competing interests exist.

**Abbreviations:** Ab, antibody; APC, antigen-presenting cell; bCII, bovine type II collagen; CD, celiac disease; CDR3, complementarity-determining region 3; CIA, collagen-induced arthritis; CII, type II collagen; CLIP, class II invariant chain peptide; CMP, collagen mimetic peptide; CNS, central nervous system; DM, HLA-DM; DO, HLA-DO; DR1, HLA-DR1; EAE, experimental autoimmune encephalomyelitis; FCS, flow cytometry; FoxP3, forkhead box P3; GWAS, genome-wide association study; HA, hemagglutinin; HCV, hepatitis C virus; HLA-DR1, human leukocyte antigen DR1; HLA-DQ2.5/8, human leukocyte antigen DQ2.5/8; IEDB, Immune Epitope Database; IP, intraperitoneally; IV, intravenously; KO, knockout; LN, lymph node; MBP, myelin basic protein; MHC, major histocompatibility complex; MHCII, major histocompatibility complex class II; MOG, myelin oligodendrocyte glycoprotein; MS, multiple sclerosis; mTEC, medullary thymic epithelial cell; MLR, mixed lymphocyte reaction; NIRF, near-infrared fluorescence; PF, precursor frequency; pMHCII, peptide-MHCII; RA, rheumatoid arthritis; SNP, single nucleotide polymorphism; TCR, T-cell receptor; Treg, regulatory T cell; T1D, type 1 diabetes; WT, wild-type.

of MHC class II epitopes expressed on the cell membrane of antigen-presenting cells (APCs). DM is known as the class II "peptide editor" because of its roles in removing class II invariant chain peptide (CLIP) and promoting immunodominant epitope selection [4,5]. Similar to DM, DO is also an α/β heterodimer that does not bind peptides, but unlike DM, which first appeared in amphibians, DO was found only in warm-blooded mammals [6] and has restricted expression to thymic medulla, B cells [7–9], and certain dendritic cell subsets [10–12]. This limited tissue distribution, late evolutionary appearance, and regulated expression of DO suggests that DO might have important regulatory effects on the selection of epitopes presented by MHC II during thymic deletion and B cell antigen presentation. Consistent with this notion, several studies, including recent genome-wide association studies (GWAS), have suggested correlations between single nucleotide polymorphisms (SNPs) in DO genes and autoimmune diseases, HCV infection, and cancer [13–18], suggesting that this MHC II accessory molecule might play an important role in the development of these pathologies.

Despite much effort, however, teasing out the mechanism of DO function both in vitro and in vivo has been challenging and debated [19–21]. In one camp [22–24], it is believed that DO functions as an inhibitor of DM, based on data obtained from overexpression or deletion of DO in cell lines, and a structural study showing that the DO–DM interface is the same as that of DM with DR1 [25,26]. A counter proposal addressing how DO functions is based on the observations that DO has differential effects on the selection of peptides, which is dependent on both peptide sequence and DM sensitivity [21,27,28]. Based on this model, DO and DM work synergistically towards the selection of immunodominant epitopes in antigen-processing compartments. While both models suggest that DO can impact the abundance of peptide-MHC (pMHCII) complexes, evidence connecting either proposed mechanisms to understanding how DO contributes to the selection of peptide repertoires by MHC II for the process of thymic selection, and to development of the various diseases mentioned above in an in vivo setting have been missing.

Since removal of autoreactive T cells occurs in the thymic medulla, we hypothesized that DO helps in shaping the CD4 T-cell repertoires by modulating the abundance of pMHCII complexes on the medullary thymic epithelial cells (mTECs). Accordingly, a lower density of pMHCII presentation in the absence of DO function could change the strength of the interactions between developing CD4 T cells and mTECs, resulting in less effective negative selection [29]. As a result, in the absence of DO, a different CD4 T-cell repertoire might be established that could include autoreactive CD4 T-cell precursors. Meanwhile, absence of DO in the periphery could also impact the presentation of certain self- or pathogenic epitopes by B cells and DCs, leading to higher susceptibility to disease development. Here, we report that lack of DO makes DO–knockout (DO-KO) mice more susceptible to the development of autoimmune diseases. These findings support a model of DO function in which DO works hand in hand with DM in both thymus and peripheral lymphoid organs to prevent autoimmunity.

## Results

### DO-KO B cells present peptides of lower kinetic stability

Peptide elution experiments utilizing purified B cells from DO-KO and DO-wild-type (DO-WT) mice were performed. We justified the use of B cells instead of mTECs for two main reasons: (1) the technical limitation of obtaining enough mTECS to attain reliable results and (2) the known similarities in expression of MHC class II molecules, DM, and cathepsins, the main components of the antigen processing machinery in mTECs [30–35], along with DO [8,36]. Analyses of two independent experiments (each comprising pools of 5 or 10 mice per group) showed that nearly 90% of eluted peptides were shared between the two strains, and 10% were

unique peptides. Categorization of the unique peptides revealed that 8.4% were from DO-WT B cells, while only 2.6% were eluted from DO-KO B cells (**Fig 1A**). To characterize the eluted peptides for affinity comparison, the Immune Epitope Database (IEDB) algorithm (http:// www.iedb.org) [37,38] was applied, which revealed that, globally, peptides from DO-WT B cells were predicted to be of higher affinity for binding to I-A$^b$ than peptides eluted from DO-KO B cells (**Fig 1B** top). This trend was even more evident when we compared just the unique peptide sequences summarized in **Fig 1B** (**bottom**). A detailed sequence, frequency of detection by mass spectrometry, and predicted affinity scores of the detected peptides are shown in **S1**–**S3 Tables**. **S1 Table** depicts all the shared peptides, where those peptides detected at higher frequencies in DO-WT are colored in periwinkle, and peptides eluted at higher frequencies in DO-KO mice are colored light green. As shown, while some of the highest-affinity peptides are similarly distributed, the majority of DO-WT–derived peptides fall at the top of the ranking of affinity scores, remaining above an arbitrarily chosen affinity of approximately 11.00, while DO-KO eluted peptides are placed at the lower part of the table. When the same strategy was applied to the DO-WT unique peptides (**S2 Table**), 32 peptides had a score lower than 11, and only 15 came out higher than 11. In **S3 Table**, peptides unique to DO-KO mice are tabulated similarly, showing a complete switch, with only 8 peptides having scores higher than approximately 11 versus 30 peptides scoring lower than 11. Because of the finding that peptides eluted in the absence of DO appear to be of lower affinity, it is likely that the observed lower percentage of unique DO-KO–derived peptides might be due to lower abundance, causing a disproportionate loss during sample preparations, hence failure of mass spectrometry to detect them. Indeed, the results shown in **S1 Fig** (S1 Data) corroborates with this interpretation, because when B cells from only five DO-WT mice were used, the number of detected peptides was approximately 1,250 but increased to 1,660 when 10 mice were used for the elution studies, a difference of 410 peptides. The numbers of peptides eluted form DO-KO mice, however, increased from 944 to 1,290 when 5 versus 10 mice were used (a difference of 246 peptides), with the majority of peptides eluted from DO-KO mice falling in the predicted affinity scores lower than approximately 11.

## DO-KO CD4 T cells are autoreactive

Finding that differences existed in the peptide repertoires of DO-WT and DO-KO mice, we next interrogated if this alteration would indeed affect the CD4 T-cell repertoire. We therefore evaluated the total percent and numbers of splenic CD4 T cells in naïve DO-WT and DO-KO mice. Interestingly, the spleens of naïve DO-KO mice generally contained larger numbers of total cells; however, only a slight proportion of them were CD4$^+$ T cells (**Fig 2A**). When CD4$^+$ T cells were further analyzed for the expression of CD25, naïve DO-KO mice harbored a significantly larger proportion of CD4$^+$CD25$^+$ T cells (**Fig 2B**). As CD25 expression is a common marker for regulatory T cells (Tregs), we also looked at the expression level of the forkhead box P3 (Foxp3) transcription factor in naïve DO-KO mice and found elevated levels of Tregs (**Fig 2C**). However, when we tested the ability of DO-KO Tregs to suppress activated CD4 T cells in an in vitro suppression assay, both DO-WT and DO-KO Tregs performed similarly (**S2 Fig** and **S9 Data**).

Because only slight alterations in the total numbers of CD4 T cells were found in naïve DO-KO mice, we next performed a modified mixed lymphocyte reaction (MLR) experiment called "prime and restimulate" [39]. This strategy is expected to boost the frequency of CD4 T cells from either DO-WT or DO-KO mice reacting against the peptide repertoires presented by B cells of the other strain. Briefly, DO-WT mice were immunized intraperitoneally (IP) with DO-KO splenocytes and 9 days later, total DO-WT splenocytes were harvested and used

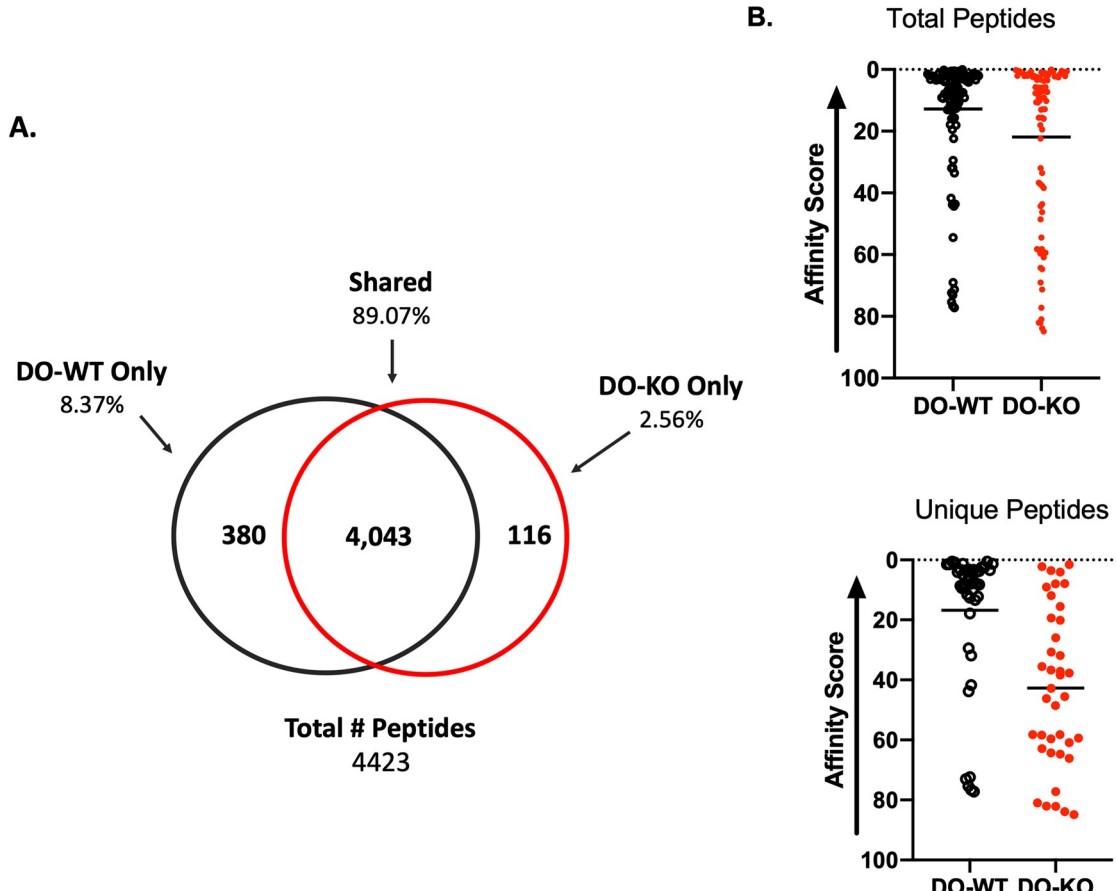

**Fig 1. Peptides eluted from I-A$^b$ in DO-KO are largely predicted to be poor binders.** (A) Breakdown of pooled replicate elution experiments. Sample information was pooled and eluted peptides divided into three categories: (1) peptides only in DO-WT samples, (2) peptides only in DO-KO samples, and (3) peptides shared between both samples. (B) Breakdown of peptides based upon the predicted IEDB affinity score. The closer to zero, the better the predicted affinity (i.e., better binding). Both total peptides (top) and unique peptides (bottom) were subdivided. Line represents the mean score. Data are representative of 2 replicate experiments. Experimental results depicted in this figure can be found in S1 Data. DO, HLA-DO; KO, knockout; IEDB, Immune Epitope Database; WT, wild-type.

as responder cells. Responder cells were then mixed with purified DO-KO B cells in an in vitro restimulation culture for a week. Cultured cells were then labeled with a proliferation dye before restimulation with a second round of naïve B cells, and were evaluated for the extent of T-cell proliferation. As shown, nearly 60% of DO-WT CD4 T cells proliferated to DO-KO B cell stimulation after two rounds of in vitro restimulation (**Fig 3A right**). In contrast, we observed only approximately 3% proliferation in the control samples showing response of DO-WT CD4 T cells to autologous B cell stimulation (**Fig 3B right**). The response pattern of stimulated DO-KO CD4 T cells, however, was strikingly different. In response to DO-WT stimulation DO-KO CD4 T cells proliferated poorly, averaging approximately 18% (**Fig 3A left**), which was surprisingly the same as their proliferation in response to autologous DO-KO B cell stimulation (**Fig 3B left**). We also looked to assess the precursor frequency (PF), or percent of dividing cells within each stimulating condition [40]. To do this, we used the cell track function of the ModFit LT 5.0 analysis software. The PF represents the proportion of CD4 T cells that have undergone at least one division due to stimulation by the opposite strain. Similar to what was observed in **Fig 3A,** DO-KO stimulated DO-WT cultures had a higher PF of

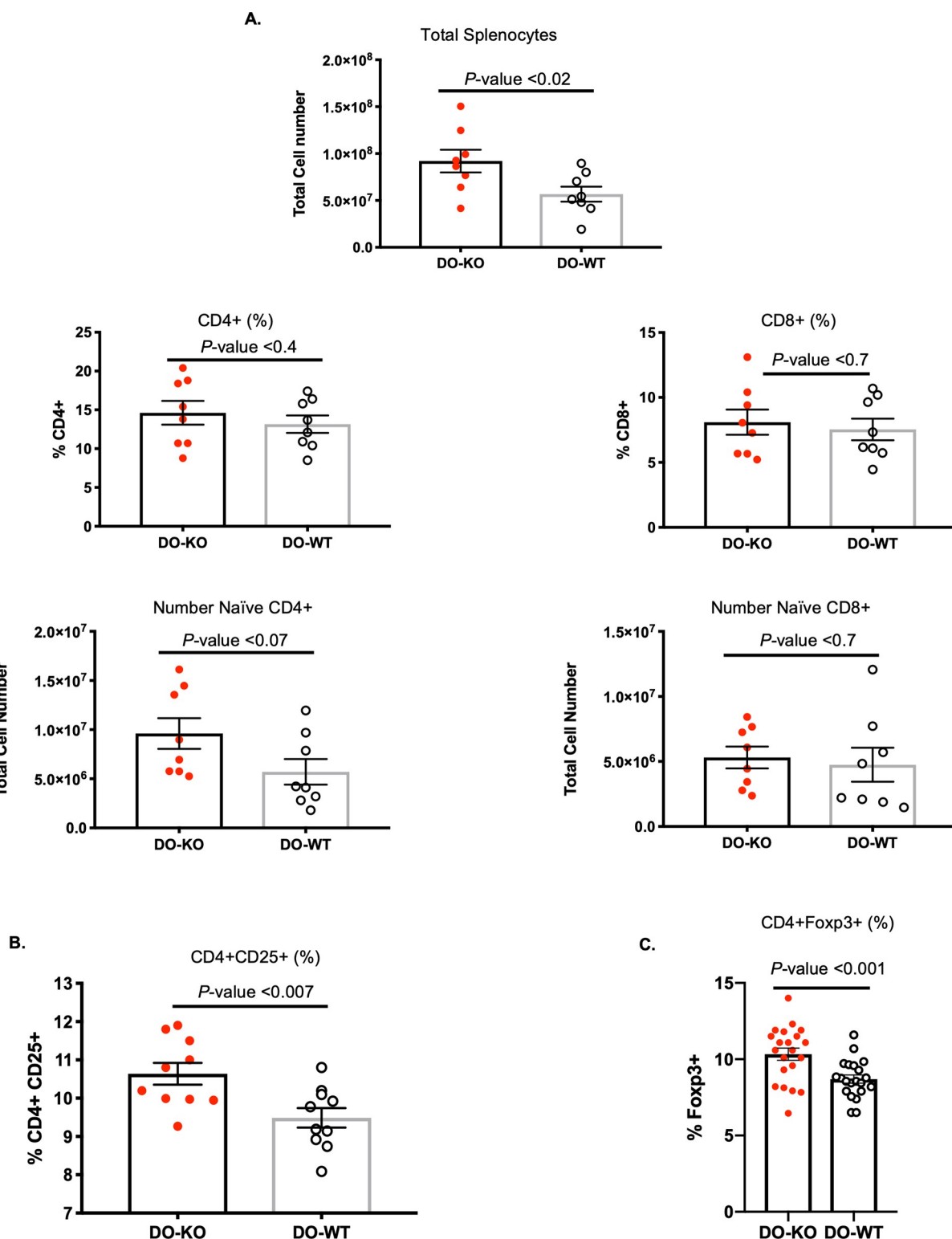

**Fig 2. Naïve DO-KO mice have an expanded CD4 T-cell compartment as well as increased expression of Tregs.** (A) Breakdown of T-cell compartment in the spleens of naïve DO-WT (white) and DO-KO (red) mice. $N = 8$ mice per group. Data are represented as mean ± SEM. (B). Subdivision of the CD4 T-cell compartment showed that DO-KO mice have increased percentage of $CD4^+CD25^+$–expressing cells. $N = 8$ mice per group. Data are represented as mean ± SEM. (C). Naïve expression of the Foxp3 transcription factor in naïve $CD4^+$ T cells from DO-WT (white) and DO-KO (red) mice. $N = 21$ mice per group. Experimental results depicted in this figure can be found in S2 Data. Data are represented as mean ± SEM. DO, HLA-DO; Foxp3, forkhead box P3; KO, knockout; Treg, regulatory T cell; WT, wild-type.

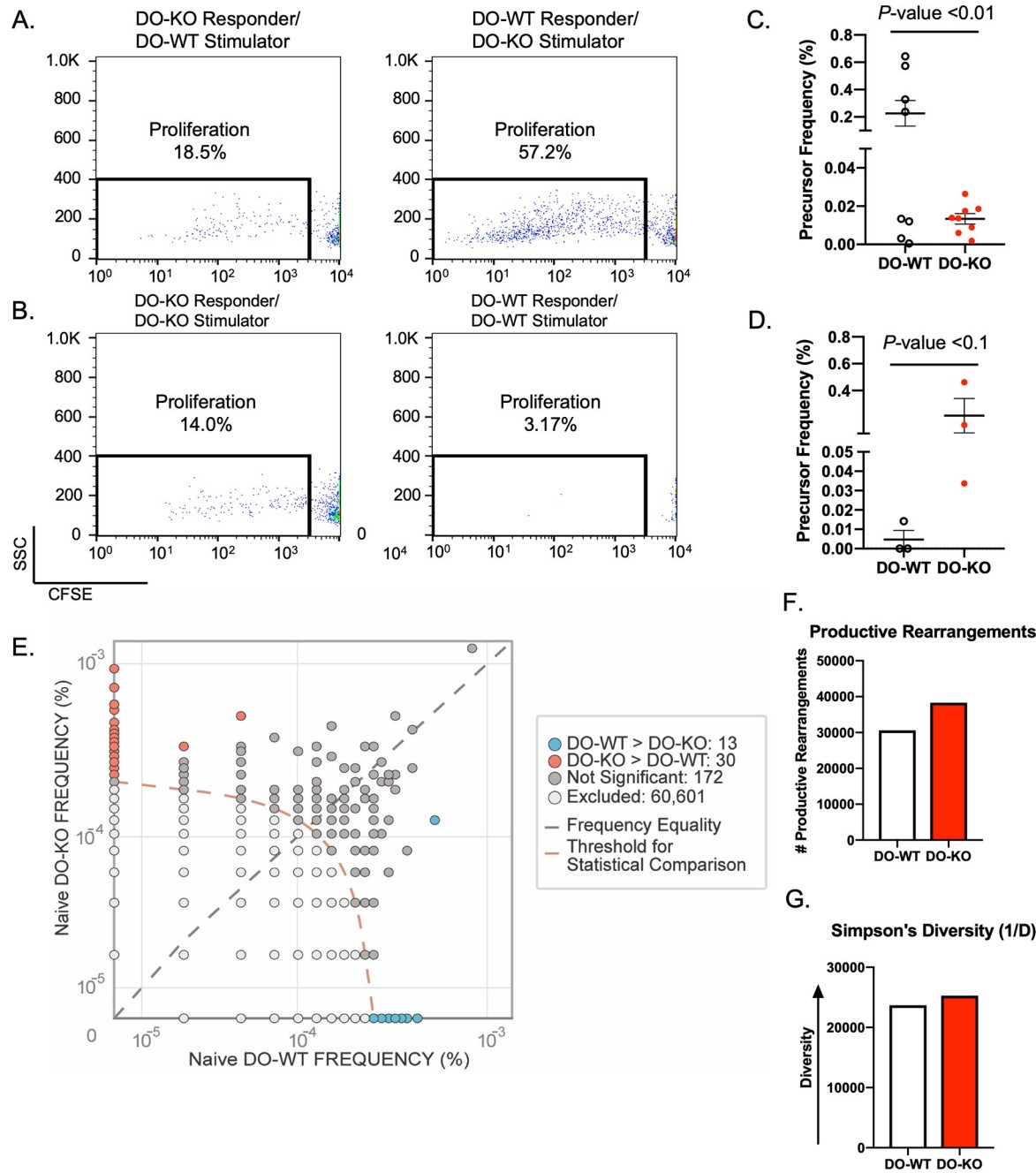

**Fig 3. Naïve DO-KO mice have a wider T-cell repertoire and are autoreactive.** (A) Representative flow plots showing the proliferation of responder CD4+ T cells after stimulation by heterologous B cells. (B) Representative flow plots showing the proliferation of responder CD4+ T cells after stimulation by autologous B cells. (C) PF of responding antigen specific CD4+ T cells in either DO-WT (open) or DO-KO (red) in the modified MLR experiments. $N = 8$ mice/group (D) PF of responding antigen-specific CD4+ T cells in either DO-WT (open) or DO-KO (red) after autologous B cell stimulation in the modified MLR experiments. $N = 3$ mice/group. Data are represented as mean ± SEM. (E) TCR β CDR3 sequence comparison of naïve CD4+ T cells as determined by next generation sequencing. Dot plot shows the number of TCR β chain sequences that were highly abundant (>10 copies/sample, $P$ value < 0.01) in either naïve DO-KO (red) or DO-WT (blue) samples. Data are representative of 5 mice pooled per genotype. (F) Total number of productive rearrangements identified in DO-WT (white) or DO-KO (red) samples. (G) Simpson's Diversity, a measure of species richness or abundance. Experimental results depicted in this figure can be found in S3 Data. CDR3, complementarity-determining region 3; MLR, mixed lymphocyte reaction; PF, precursor frequency; TCR, T-cell receptor.

antigen-specific CD4 T cells than DO-KO cultures (Fig 3C). Modeling of the autologously stimulated cultures further confirmed the observations in Fig 3B that DO-KO mice had a higher PF than DO-WT mice (Fig 3D). Taken together, these data further suggest that (a) the peptide repertoire presented by DO-KO B cells is different from DO-WT and (b) the DO-KO CD4 T-cell repertoire likely includes autoreactive cells.

To better define the TCR repertoire of naïve DO-KO mice (x-axis) (Fig 3E), deep sequencing of TCR β chain complementarity-determining region 3 (CDR3) was performed on sorted CD4$^+$ CD3$^+$ T cells and compared to the TCR β chain repertoire of naïve DO-WT (y-axis). Analysis of the pooled samples found that DO-KO mice had a higher number of productive rearrangements (i.e., 35,863 DO-KO versus 28,884 DO-WT) (Fig 3F), consistent with the hypothesis that the lack of DO leads to altered thymic deletion. Further comparison showed that while the majority of TCR β clones were shared between the DO-KO and DO-WT samples, 30 TCR β clones were highly enriched (>10 copies, P value < 0.01) in the naïve DO-KO sample, while only 18 TCR β clones were highly enriched (>10 copies, P value < 0.01) in the naïve DO-WT sample (Fig 3E). The Simpson's Diversity calculation (1/D), in which numbers much larger than 1 are indicative of polyclonality (Fig 3G), shows that both samples were equally polyclonal in the naïve state, although the DO-KO sample was slightly more polyclonal. This slight increase in diversity of the DO-KO–derived clonal population supports the need for a "prime-restimulate" strategy.

## Naive DR1$^+$DO-KO mice and DR1$^+$DO-WT mice have similar collagen-specific CD4 T-cell precursors frequencies

If more autoreactive clones exist within the DO-KO mice, we next asked if any could be identified. Previous work, using the non-obese diabetic (NOD) mouse model of spontaneous diabetes development showed that overexpression of DO in CD11c$^+$ DCs protected DO-KO mice from developing diabetes [19]. In contrast, recent GWAS have reported associations between SNPs in DO genes and multiple maladies, including increased susceptibility to RA [13,14]. Collagen-induced arthritis (CIA) is a murine model of human RA [41] that can be induced by immunization with type II collagen (CII) protein in DR1 transgenic mice [42]. Although the immunodominant epitope of CII, CII(280–294), forms highly stable complexes with DR1 (t$_{1/2}$ = 180 hours), the complex is DM-sensitive [43]. To be able to test susceptibility to CIA, we generated DR1$^+$DO-KO and DR1$^+$DO-WT mice by backcrossing DR1 transgenic mice to DO-KO C57BL/6 mice for over 10 generations followed by inbreeding to homozygosity. Both DR1$^+$DO-KO and DR1$^+$DO-WT mice express a chimeric human/mouse DR1/I-E$^b$ molecules, in which the peptide binding groove is DR1, but the membrane proximal domains of MHC II is murine I-E$^b$, to allow full interaction of murine CD4 molecules with I-E$^b$ [42]. DR1-expressing mice were also backcrossed to WT C57BL/6 mice to create DR1$^+$DO-WT control mice. Characterization of naïve DR1$^+$DO-WT and DR1$^+$DO-KO mice showed no detectable changes in any lymphoid compartments (S3 Fig). Moreover, these mice have been monitored for more than 2 years without showing signs of spontaneous autoimmune diseases. To evaluate levels of CII(280–294)-specific CD4 T cells in the peripheral lymphoid organs of naïve DR1$^+$DO-WT and DR1$^+$DO-KO mice, a tetramer enrichment strategy using a CII(280–294)/DR1 tetramer was employed (**Materials and methods**). After analysis, no statistically significant difference in CII(280–294)-specific CD4 T-cell precursor frequencies was noted between DR1$^+$DO-WT mice and DR1$^+$DO-KO mice (Fig 4A). These observations suggest that the thymic deletion of CII(280–294)-specific CD4 T cells in naïve DR1$^+$DO-KO mice was not impaired in the absence of DO. To evaluate the peripheral presentation of the CII(280–294) epitope, we immunized DR1$^+$DO-WT and DR1$^+$DO-KO mice with CII protein and looked at

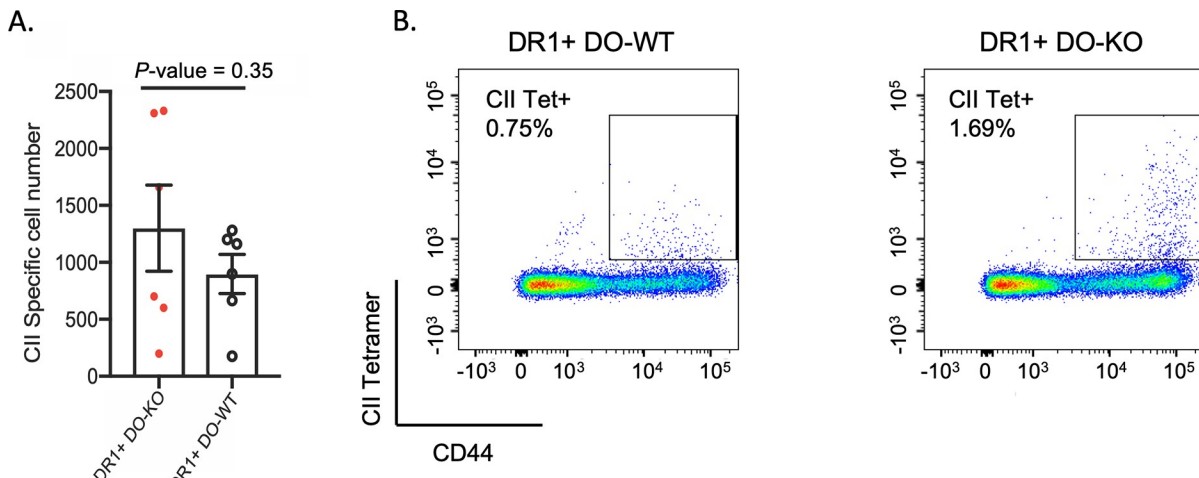

**Fig 4. Naive DR1⁺DO-KO mice and DR1⁺DO-WT mice have similar collagen-specific CD4 T-cells precursor frequencies.** (A) Total number of CII(280–294)-specific CD4 T cells in DR1⁺DO-KO (red) and DR1⁺DO-WT (white); $N = 7$. Data are represented as mean ± SEM. (B) Increased frequencies of CII(280–294)/DR1 tetramer positive CD4 T cells directly isolated from CII protein immunized mice 9 days prior to cell isolation indicate enhanced in vivo processing and presentation of the immunodominant CII(280–294) epitope by DR1 in DO-KO mice. Experimental results depicted in this figure can be found in S4 Data. CII, type II collagen; DR1⁺, HLA-DR1; DO, H2-O; KO, knockout; Tet, tetramer; WT, wild-type.

the frequencies of freshly isolated CII(280–294)-specific CD4 T cells from the draining lymph nodes (**Fig 4B**). This better expansion of CII(280–294)-specific T cells in DO-KO mice is a positive indication of CII processing and presentation of its immunodominant epitope by APCs in vivo.

## DR1⁺ DO-KO mice are more susceptible to the development of RA

We next induced CIA in DR1⁺DO-WT and DR1⁺DO-KO mice using a modified CIA protocol [42] and assessed the development of CIA by a novel whole-body near-infrared fluorescence (NIRF) imaging technique [44]. Briefly, DR1⁺DO-WT and DR1⁺DO-KO mice received a primary subcutaneous tail injection of bovine type II collagen (bCII) protein followed by a second subcutaneous tail immunization 21 days later (**Materials and methods**). It is well-known that manifestation of CIA disease is due to collagen denaturation initiated by immune cells [45]. Recently, a caged collagen mimetic peptide (CMP) probe that specifically binds to denatured collagen actively undergoing degradation has been defined [46–48]. This useful tool has further been developed for tracking collagen remodeling activity in vivo [49–52]. Of particular interest is that alongside the labeled CMP probe, additional probes conjugated to antibodies specific to different cell markers, such as CD4 or labeled tetramers, can be used simultaneously. We therefore attempted to assess disease development using the CMP probe labeled with an infrared dye (IRDye680RD) together with an anti-CD4 antibody labeled with a second spectrally distinct infrared dye. The CMP probe was injected intravenously (IV) into the tail of the diseased mice, followed by a single IP injection of anti-CD4 antibody probe to detect if any co-localization between CD4 T cells and remodeled collagen had occurred. In order to obtain high-resolution in vivo images, diseased mice were skinned. Image analyses showed striking differences in co-localization of CD4 and CMP probes near the affected joints (**Fig 5A**). Co-localization of CD4 T cells (green) with the CMP probe (red) was only found in the joints of diseased DR1⁺DO-KO mice (**Fig 5A bottom**) as compared to DR1⁺DO-WT mice (**Fig 5A top**). To further explore the specificity of the CD4 cells near the affected joints, in two separate

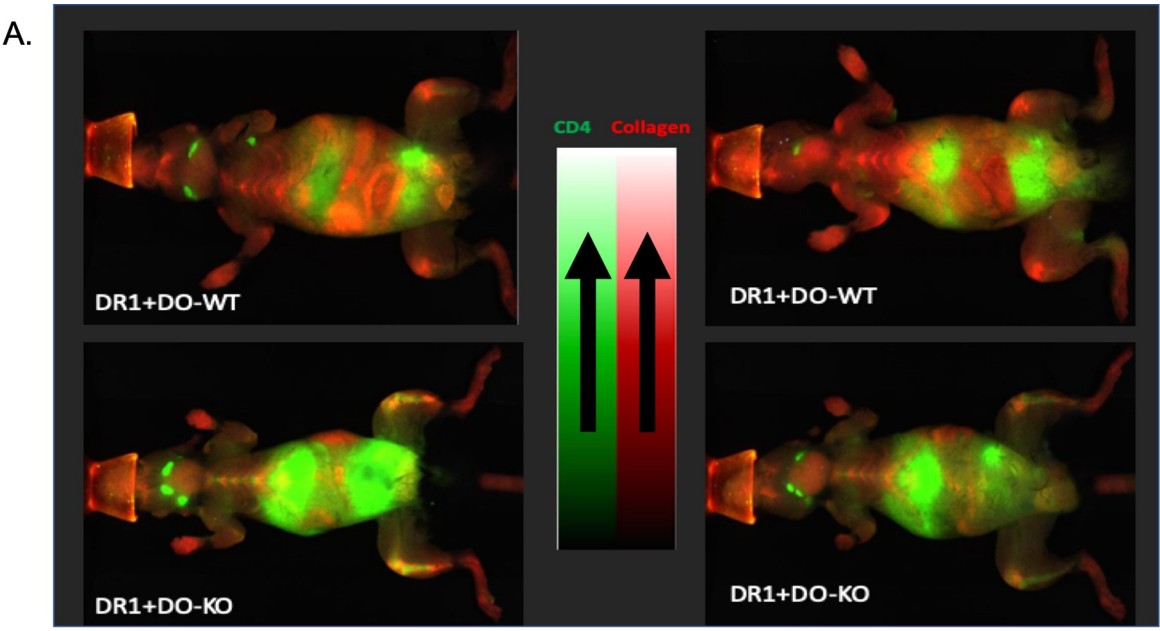

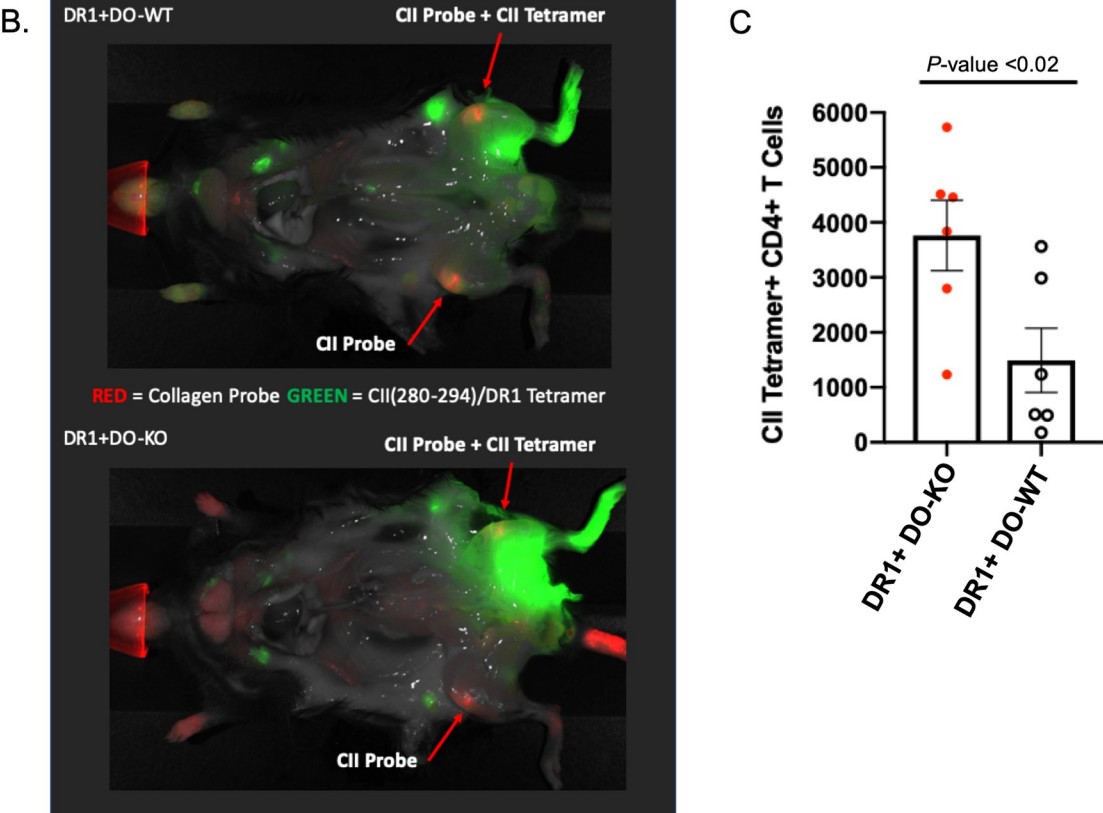

**Fig 5. CIA diseased DR1⁺DO-KO mice showed increased co-localization of CII(280–294)/DR1 tetramer positive CD4 T cells with denatured collagen.** (A) Representative images showing whole-body NIRF imaging of CMP probe (red) injected (IV) followed by CD4 probe injected IP (green) 48 hours prior to imaging. Co-localization of denatured collagen with CD4 infiltration is denoted by orange signal. Representative of 3 independent experiments. (B) Representative images of CIA diseased DR1⁺DO-KO (top) and DR1⁺DO-WT (bottom) mice showing the distribution of in vivo stained CII(280–294)/DR1 tetramer positive T cells (green) 72 hours after footpad injection. CMP probe (red) was injected IV at the same time. Representative of 2 independent experiments. (C) Increased number of in

vivo labeled CII(280–294)/DR1 tetramer-specific CD4 T cells were recovered from the popliteal lymph nodes of diseased DR1$^+$DO-KO (red) or DR1$^+$DO-WT (white). Data represented as mean ± SEM of 6 individual mice in two separate experiments. Experimental results depicted in this figure can be found in S5 Data. CIA, collagen-induced arthritis; CII, type II collagen; CMP, collagen mimetic peptide; DR1$^+$, HLA-DR1; DO, H2-O; KO, knockout; IP, intraperitoneally; IV, intravenously; NIRF, near-infrared fluorescence; WT, wild-type.

experiments, diseased mice were injected intra-footpad with CII(280–294)/DR1 tetramers. Mice were imaged 72 hours post injection to assess the total number of in vivo labeled CII-specific CD4 T cells in either DR1$^+$DO-WT or DR1$^+$DO-KO mice (Fig 5B). Ex vivo antibody staining of cells from popliteal lymph nodes (LNs) of mice imaged in Fig 5B showed that diseased DR1$^+$DO-KO mice had an increased number of CII(280–294)-specific CD4 T cells (Fig 5C). Data in Figs 4 and 5 suggest that CIA development did not correlate with the PF of CII-specific CD4 T cells in DR1$^+$DO-KO and DR1$^+$DO-WT mice, but rather with increased presentation of the DM-sensitive CII(280–294) epitope in peripheral tissues due to loss of DO function.

## Naïve DO-KO mice have higher CD4 T-cell precursor frequencies for myelin oligodendrocyte glycoprotein

Next, we chose to examine another antigen known to induce an autoimmune disease in mice. Myelin oligodendrocyte glycoprotein (MOG) is a minor component of the myelin sheath that is targeted by T cells and autoantibodies in autoimmune diseases of the central nervous system (CNS). Looking at the immunodominant epitope for I-A$^b$, MOG(38–51), we expected it to be DM-resistant based on (a) data showing the likelihood of a tyrosine (Y40) residue fitting into the I-A$^b$ P1 pocket [53] and (b) molecular modeling studies [54], which showed striking similarities to the DM-resistant hemagglutinin (HA)(306–318)/DR1 complex [55]. The work of Poluektov and colleagues predicts that a higher number of DM-resistant MOG epitopes will be presented in the thymic medulla of DO-WT mice, leading to efficient deletion of MOG-specific CD4 T cells (S4 Fig). Conversely, in the absence of DO, more MOG-specific CD4 T cells would possibly be found in the periphery of DO-KO mice. Similar to the measurement of CII(280–294)-specific CD4 T-cell PF, a MOG(38–51)/I-A$^b$ tetramer was employed to enumerate MOG(38–51)-specific CD4 T cells in naïve mice. In accord with this hypothesis, naïve DO-KO mice showed higher numbers of MOG(38–51)-specific CD4 T cells as compared to DO-WT mice (Fig 6A). Because of a wider mouse-to-mouse variability in naïve DO-KO mice, we also performed an in vivo boost of the MOG(38–51) specificity through peptide immunization and ex vivo culture. Confirming the trend that we saw in naïve mice, DO-KO mice showed an increased percentage of MOG-specific CD4 T cells (Fig 6B).

## DO-KO mice are more susceptible to experimental autoimmune encephalomyelitis development

To investigate if increased MOG specific precursor frequencies in DO-KO mice correlated directly with autoimmune disease development, we tested the development of experimental autoimmune encephalomyelitis (EAE), a mouse model of multiple sclerosis (MS) (**Materials and methods**). Because MOG is a component of the myelin sheath, which both protects nerves and aids in transmitting neuronal signals, destruction of the myelin sheath (demyelination) leads to interruptions in signal transmission and ultimately paralysis. Immunized mice were therefore visually monitored for the development of neurological symptoms (i.e., paralysis), beginning with a limp tail (Score 1) and progressing to full-body paralysis (Score 4). From these studies, we found that DO-KO mice had an accelerated onset of disease compared to DO-WT mice (Fig 7A). Disease symptoms (Score 1) appeared around Day 8–10 and quickly

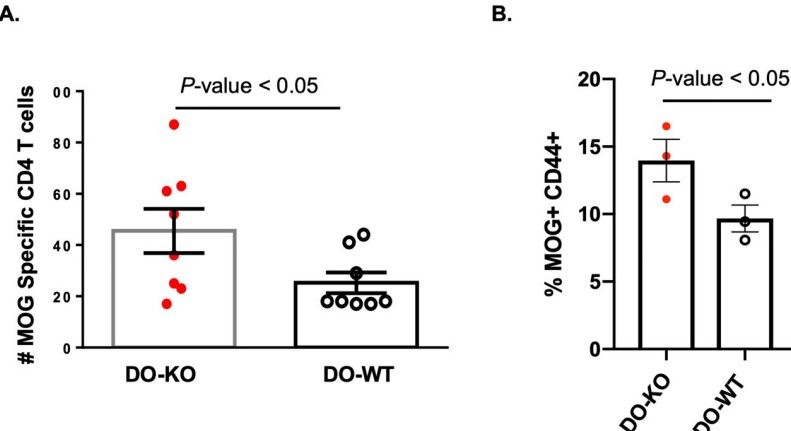

**Fig 6. Naïve DO-KO mice have higher levels of MOG-specific CD4 T cells.** (A) Total numbers of MOG-specific CD4[+] T cells in naïve DO-KO (red) and DO-WT (white) mice; *N* = 8. Data represented as mean ± SEM. (B) In vitro boosting with MOG in DO-KO and DO-WT showed an increased percentage of MOG-specific CD4 T cells in the DO-KO mice. Tetramer positive gate is from CD4[+] T-cell gate. Each dot is an individual mouse, *N* = 3 per group. Experimental results depicted in this figure can be found in S6 Data. DO, H2-O; KO, knockout; MOG, myelin oligodendrocyte glycoprotein; WT, wild-type.

progressed to advanced disease (Score 3–4) by Day 14–16 in DO-KO. In contrast, DO-WT mice started showing symptoms around Day 12 and progressed to advanced disease scores by Day 20. Total cell infiltration into the CNS tissue was slightly higher in DO-KO mice, but no change in total brain weight was observed (**S5 Fig**). To further correlate the state of disease with CD4 infiltration, we performed in vivo NIRF whole-body imaging on diseased DO-WT and DO-KO mice using an antibody (Ab) probe specific to myelin basic protein (MBP). The Ab reacts with MBP only when the myelinated glia cells are damaged during disease development [56]. Thus, by detecting demyelination, whole-body imaging allowed us to fully visualize the co-localization of CD4 T cells at the sites of demyelination occurring in diseased mice. Interestingly, when mice of various disease scores were imaged, we found increased co-localization of infiltrating CD4 T cells with anti-MBP staining in DO-KO mice, but not in DO-WT mice (**Fig 7B**). These data not only confirmed the flow cytometric findings that diseased DO-KO mice have a greater influx of lymphocytes into their CNS tissue (**S5 Fig**), it also verified the massive demyelination that occurs during the disease.

To find out if the higher PF found in DO-KO mice correlated with disease development, we used a MOG(38–51)/I-A[b] tetramer to enumerate MOG-specific CD4 T cells directly isolated from CNS tissues of diseased DO-KO and DO-WT mice. The results shown in **Fig 7C** indicate recovery of a larger percentage, as well as total numbers of MOG(38–51)-specific CD4 T cells from DO-KO CNS. This difference was even greater when looking specifically at Day 10–12 of disease development (**Fig 7D**). These findings showed that EAE development in DO-KO mice did correlate with the precursor frequencies of MOG(38–51)-specific CD4 T cells in naïve and disease states.

To evaluate differences in the presentation of naturally processed MOG(38–51)/I-A[b] by APCs from the CNS in the absence of DO, we used 2D2 transgenic CD4 T cells specific for MOG(38–51)/I-A[b] as a highly sensitive readout system [57]. The 2D2 transgenic T cells were cocultured with various concentrations of harvested CNS infiltrating APCs (Day 17) from EAE-diseased DO-WT and DO-KO mice. **Fig 8A** represents three technical replicates of pooled samples from five individual diseased mice. As controls, APCs from the draining LNs of diseased mice were isolated and cocultured with 2D2 CD4 T cells. We found that DO-WT

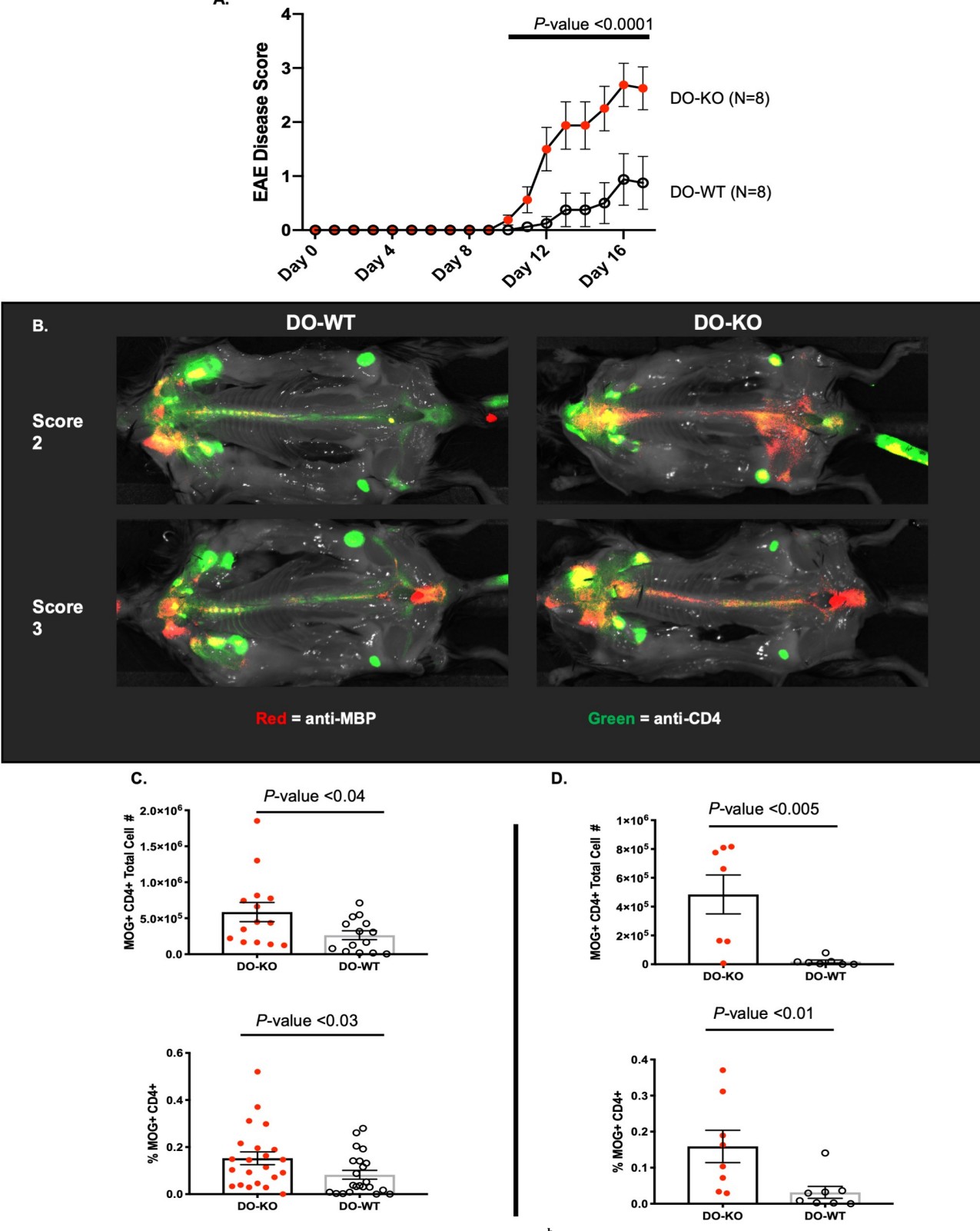

**Fig 7. Accelerated EAE disease onset and increased recovery of MOG(38–51)/I-A$^b$ tetramer positive CD4 T cells from the CNS tissue of DO-KO mice.** (A) Representative curves showing the time course of disease development in DO-KO (red) and DO-WT mice (white). $N$ = 5 mice per group, representative of >15 repeat experiments. Score system: 0 = no symptoms, 1 = limp tail, 2 = limp tail + partial hind limb paralysis, 3 = limp tail + total

hind limb paralysis, 4 = limp tail + total hind limb paralysis + partial forelimb paralysis. Data represented as mean ± SEM. (B) Representative images showing co-localization of CD4 T cells (green) with MBP (red) in mice scoring either 2 (top) or 3 (bottom) in DO-WT or DO-KO mice. Co-localization is denoted in the merged image by orange signal. (C) Total number (top) ($N = 14$) and percent (bottom) ($N = 22$) of MOG-specific, $CD44^+$ CD4 T cells recovered from the CNS tissue, humanely killed on Day 16–18. (D) Total number (top) and percent (bottom) of MOG-specific, $CD44^+$ CD4 T cells recovered from the CNS tissue of diseased DO-KO (red) or DO-WT (white) mice, focusing specifically on the accelerated onset phase of disease seen in DO-KO (red) as compared to DO-WT (white) mice; the kill point was Day 10–12. $N = 7$ mice. Percent data has been normalized to account for varying total events collected during FC analysis. $\% \ Positive = \left( \left( \frac{\#CD4 \ MOG \ positive \ events \ collected}{Total \# events \ collected} \right) x \ 100 \right)$. Experimental results depicted in this figure can be found in S7 Data. CNS, central nervous system; DO, H2-O; EAE, experimental autoimmune encephalomyelitis; FC, flow cytometry; KO, knockout; MBP, myelin basic protein; MOG, myelin oligodendrocyte glycoprotein; WT, wild-type.

CNS APCs were more efficient in activating 2D2 cells as indicated by increased CD69 expression, likely due to a higher abundance of MOG(38–51)/I-A$^b$ being presented by CNS APCs from diseased DO-WT mice (**Fig 8A and 8B**). The control samples using draining LN APCs from either DO-WT or DO-KO mice did not induce up-regulation of CD69 (**Fig 8C**). An important control shown in **S6 Fig** indicates that both DO-KO and DO-WT naïve B cells were equally able to present MOG(38–51)/I-A$^b$ to 2D2 T cells when pulsed with the peptide. Hence, there is no difference in the level of I-A$^b$ on B cells between naïve DO-KO and DO-WT mice. These data suggest that while presentation of MOG(38–51)/I-A$^b$ is more robust in CNS APCs from DO-WT mice, the more severe disease seen in the DO-KO mice is likely due to higher PF of the MOG(38–51)-specific T cells. In summary, these findings suggest that DO can impact both the selection of the T-cell repertoire as well as peripheral antigen presentation to prevent development of autoimmunity.

## Discussion

Autoimmune diseases affect more than 23.5 million Americans, a number that keeps rising each year. As such, major research efforts have been undertaken to understand what risk factors contribute to disease development. Because GWAS have linked susceptibility to the development of autoimmune diseases and SNPs in DO, and because DO is selectively expressed in the thymic medulla [8], where autoreactive CD4 and CD8 T cells undergo negative selection, loss of proper epitope presentation in DO-KO mice would likely result in altered CD4 T-cell selection. We sought to clarify the role of DO in the regulation of CD4 T-cell development and peripheral antigen presentation in vivo. To date, two competing theories exist as to how DO affects the levels of pMHC complexes; in one, DO has evolved to simply inhibit the function of DM [58,59]. In another, DO works cooperatively with DM, thereby ensuring effective epitope selection based on the nature of the epitope [21,27,28,60,61]. But systematic in vivo studies using DO-KO mice and supportive in vivo evidence for either model have been missing.

In this report, using a multiplicity of approaches including (i) peptide elution studies from naïve DO-KO and DO-WT B cells, (ii) a modified MLR to detect immunogenic pMHCII, (iii) TCR deep sequencing of naïve DO-KO and DO-WT CD4 T cells, (iv) comparison of CD4 T-cell precursors for MOG and CII, and finally, (v) susceptibility to CIA and EAE, we provide a systematic documentation of biological consequences resulting from the loss of DO in vivo. All these approaches support the model that DO works cooperatively with DM for prevention of autoimmune disease. In addition, we have interrogated the naturally occurring Treg populations [62] and found an increased percentage of Tregs in naïve DO-KO relative to DO-WT mice, although Tregs from DO-KO mice did not perform any better than DO-WT Tregs in the standard in vitro suppression assay.

Evaluation of differences in the eluted peptide repertoires generated in B cells from mice that did or did not express DO using IEDB affinity prediction algorithms [38,63] showed that DO-WT B cells presented a large pool of unique peptides with binding registers that were

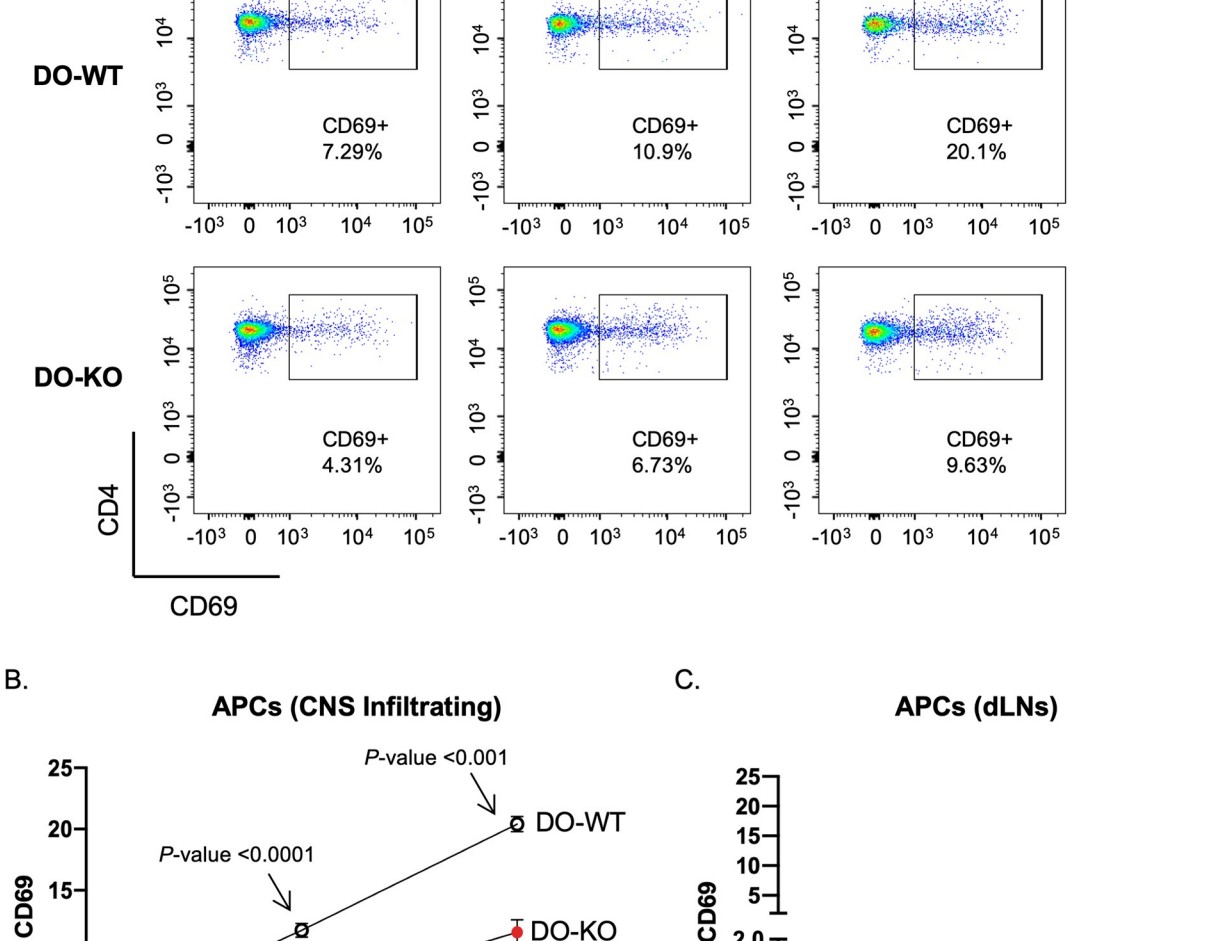

**Fig 8. CNS infiltrating APCs from DO-KO mice are less stimulatory to 2D2 T cells.** (A) Representative FCS plots showing one of the three replicates of the 2D2:APC coculture experiments. Freshly isolated 2D2 Tg CD4 T cells were cocultured with various APC concentrations $(0.25 \times 10^5, 0.5 \times 10^5,$ and $1 \times 10^5)$ isolated from the pooled CNS tissue of either DO-WT (top) or DO-KO (bottom) mice. (B) APCs isolated from the CNS of DO-KO mice present less of the MOG35–55 epitope compared to DO-WT despite having a more advanced disease phenotype. (C) Differences in presentation were not seen with APCs isolated from the dLN of the diseased mice. Data represented as mean ± SEM. $N = 4$ diseased mice pooled, each dilution plated in triplicates. Repeated in two independent experiments. Experimental results depicted in this figure can be found in S8 Data. APC, antigen-presenting cell; CNS, central nervous system; dLN, draining lymph node; DO, H2-O; FCS, flow cytometry; KO, knockout; MOG, myelin oligodendrocyte glycoprotein; Tg, transgenic; WT, wild-type.

predominantly of higher affinity than that of DO-KO eluted peptides. These findings suggest that in the absence of DO, DM is not narrowing the peptide repertoire to only those with high binding affinities. However, in a recent study using a DR1[+] B cell line, the authors showed differences in the peptide repertoires of the two cell lines with or without DO, and concluded that

peptides eluted from the DO-KO cell line were less diverse; although, their report on elution from DO-KO mice revealed only subtle differences [64]. While those observations might appear consistent with our peptide elution data, an apparent lower diversity of the eluted peptides from the DO-KO cell line could be due to a lower abundance of the class II bound peptides in the absence of DO [28]. Indeed, this postulated higher diversity of presented peptides by DO-KO B cells is supported by later experiments showing a 3-fold higher level of DO-WT CD4 T-cell proliferation to DO-KO B cells.

We next hypothesized that the absence of DO would lead to presentation of altered quantities of certain pMHCII molecules, thereby affecting thymic deletion [65]. Enumeration of CD4 T-cell precursor frequencies for immunodominant epitopes of two self-antigens, i.e., CII(280–294), known to be highly stable, yet DM-sensitive, and MOG(38–51), postulated to be DM-resistant, supported our hypothesis. DO-KO mice hold an increased PF of MOG(38–51)-specific CD4 T cells, consistent with a faulty thymic deletion. In contrast, we found an unaltered PF of CII(280–294)-specific CD4 T cells in DO-KO mice. Lack of a clear difference in self-deletion of CII(280–294)-specific T cells in periphery might reflect the high stability of the CII (280–294) epitope binding to DR1 ($t_{1/2} > 180$ hours) [43].

Clear evidence that DO plays a critical role in protection against autoimmune diseases came from two autoimmune disease models, CIA and EAE. When we tested DO-KO mice for the development of EAE, we found a more rapid onset of disease that was visually correlated with CD4 T-cell infiltration by NIRF whole-body imaging. The power of NIRF imaging used here lies in detection of demyelination by antibody labeling of the exposed MBP in CNS. Hence, by being able to detect co-localization of CD4 T cells with exposed MBP within diseased CNS tissues, we can reveal the status of the disease and its correlation with infiltrating CD4 T cells. Similarly, visualization of a worse CIA disease development in DR1$^+$DO-KO compared with DR1$^+$DO-WT mice is a powerful experimental design, as the labeled peptide mimetic is well established to be specific to denatured collagen [46,47,51,66,67].

Despite a more severe disease in both EAE and CIA in DO-KO mice, the drivers of disease development may be different. In EAE, higher PF for MOG(38–51)-specific T cells in DO-KO mice correlated well with the accelerated disease development. On the contrary, despite rather similar PF for CII(280–294)-specific CD4 T cells in DR1$^+$DO-KO and DR1$^+$DO-WT mice, DR1$^+$DO-KO exhibited more severe CII degradation. These findings can suggest that in CIA, a higher presentation of CII(280–294)/DR1 complexes in the periphery might be the main driver for the disease severity in DO-KO mice. Therefore, DO can affect presentation of both epitopes, DM-resistant MOG(38–51) and DM-sensitive CII(280–294). All of these observations provide strong evidence that loss of DO function leads to a more self-reactive CD4 T-cell repertoire, which, when activated, drives disease development. Further research, however, into the relationship between DO function and other autoimmune disorders in which the DM sensitivity of the self-epitope is known remains to be explored.

Direct and compelling evidence in support of the hypothesis that DO cooperates with DM for increased presentation of immunodominant epitopes was assessed by evaluation of MOG (38–51)/I-A$^b$ presentation by DO-WT and DO-KO APCs via a T-cell readout system. MOG-specific 2D2 CD4 T cells responded consistently at higher levels to the presented MOG(38–51)/I-A$^b$ displayed on the infiltrating DO-WT APCs directly isolated from the CNS of the diseased mice. We have provided evidence that loss of DO function generates an altered CD4 T-cell repertoire within DO-KO mice.

In conclusion, we have documented differences in the CD4 TCR repertoires shaped by DO in mice and have shown that negative selection of CD4 T cells can be controlled by DO based on the DM-sensitivity of the T-cell epitopes. These in vivo findings support the biochemical observations that DO works together with DM in order to optimize class II epitope selection

[28]. It is of note that DO has appeared later than DM in evolution. DM appeared in vertebrates that included fish, but DO evolved in warm-blooded mammals around the same time as the evolution of germinal centers [6]. Our report clearly establishes that lack of DO associates with susceptibility to autoimmune diseases, which hints that, along with the longer life spans of warm-blooded mammals, DO might have evolved for protection against the development of autoimmune diseases. Turning back to the reported correlation between SNPs in DO and autoimmune disease development in humans, with the advent of increased genetic screening and personalized medicine, it is possible that we could use the reported SNPs as diagnostic tools for stratifying patients. Being able to readily monitor people at greater risk for developing those autoimmune diseases and intervening before major deleterious symptoms have occurred could greatly improve the long-term outcome for patient health.

## Materials and methods

### Ethics statement

All animal studies were approved (M017M307) by the Johns Hopkins University Institutional Animal Care and Use committee in accordance with the Animal Welfare Act. Animals were humanely killed either by isoflurane or $CO_2$ inhalation before ex vivo studies were performed.

### Mice

C57BL/6 (B6) mice were purchased from the Jackson Laboratory (Bar Harbor, ME). $I$-$A^b$+-$H2$-$O^{-/-}$ (DO-KO) mice used in the above experiments were generated by backcrossing *129. H2-$O^{-/-}$* mice (Lars Karlsson, Johnson and Johnson Pharmaceutical Research and Development, San Diego, CA) onto B6 for 10 generations by P. Jensen and X. Chen (University of Utah) and kindly gifted to us. The original HLA-DR1 (DRB1*0101) (DR1) transgenic mice (obtained from Dr. Dennis Zaller, Merck) express a fusion MHC II molecule containing the DR1 binding groove and the membrane proximal domain of murine I–E molecule [42]. The resulting DR1 mice were backcrossed to MHC class II KO mice for 12–16 generations to eliminate endogenous class II proteins (I-$A^f$) and were then inbred to homozygosity. DR1$^+$DO-KO mice were generated by crossing the DO-KO mice with transgenic DR1 mice for >10 generations to achieve DO-KO homozygosity. The DR1$^+$DO-WT were generated by crossing the DR1 mice with B6 mice for >10 generations. All DR1$^+$H2-O mice still express murine I-$A^b$ molecules from the B6 background. All mice were housed in the Johns Hopkins University animal facilities under virus-free conditions. All experiments were performed in accordance with protocols approved by the Animal Care and Use Committee of the Johns Hopkins University School of Medicine.

### Peptides and proteins

Collagen (280–294) peptide (CAGFKGEQGPKGEPGP) and MOG (35–55) peptide (*MEVGWYRSPFSRVVHLYRNGK)* were both synthesized by Peptide 2.0 (Chantilly, VA) at >98% purity. bCII protein (Immunization Grade) was purchased from Chondrex (Redmond, WA).

### Tetramers and antibodies

PE-conjugated MOG tetramer as well as biotinylated DR1 monomers were ordered from NIH Tetramer Core Facility (Atlanta, GA). PE-conjugated DR1/Collagen II (280–294) tetramers were created in-house using the CLIP/DR1 monomers and a NIH Tetramer Core Facility protocol. Briefly, the process involved cleavage of the CLIP peptide from the DR1 monomer and an exchange reaction with the collagen II (289–294) peptide. For the flow cytometry

experiment, Brilliant Violet 421-CD44, Alexa Fluor 700-CD4, PE-Cy7-CD8, FITC-CD3, BV421-CD25, Alexa647-Foxp3, APC-B220, APC-CD11c, PE-Cy7-CD69, and APC-F4/80 from Biolegend (San Diego, CA) and fixable viability dye eFluor 780 from eBioscience (San Diego, CA) were used.

## Peptide elution

Total B cells (Elution 1: $4–5 \times 10^7$; Elution 2: $4–5 \times 10^8$) from the spleens of naïve DO-WT and DO-KO mice were magnetically isolated (StemCell Technologies, Vancouver, Canada) prior to cell lysis. Isolated cells were lysed with 1% NP-40 (50 mM Tris-HCL + 1× HALT protease and phosphatase inhibitor) for 1 hour at RT. Lysate was centrifuged at $40,000g$ to remove cell debris before immunoprecipitation of I-A$^b$ molecules. For I-A$^b$ isolation, cleared cell lysate was rotated overnight at 4˚C in the presence of an anti–I-A$^b$ antibody (clone Y-3P) coupled resin. MHC molecules were eluted from the Y-3P antibody by mild acid elution (0.1% TFA). Peptides were then isolated via 1% TFA and 40% MeOH elution. Residual detergent was removed by the Pierce Detergent Removal Spin Columns (ThermoFisher Scientific, Waltham, MA) and HILIC Column (ThermoFisher Scientific, Waltham, MA) cleanup before running on the LC-MS.

## Mass spectrometry

Cleaned peptide samples were analyzed by nano-LC/MSMS QE_Plus in FTFT using 60-minute total LC run. Tandem MS2 spectra were processed by Proteome Discoverer (v1.4 Thermo-Fisher Scientific) in three ways, using 3Nodes: common, Xtract (spectra are extracted, charge state deconvoluted, and deisotoped using Xtract option, at resolution 55 K at 400 Da), and MS2 Processor. MS/MS spectra from 3Nodes were analyzed with Mascot v.2.5.1 Matrix Science, London, United Kingdom ([www.matrixscience.com](http://www.matrixscience.com)), using 2015RefSeq database with concatenated decoy database, specifying mus_musculus species, No enzyme, precursor mass tolerance 6 ppm, fragment mass tolerance 0.02 Da and Oxidation(M), carbamidomethyl c, and DeamidationNQ as variable modifications. Scaffold (version Scaffold_4.8.9, Proteome Software, Portland, OR) was used to validate MS/MS-based peptide and protein identifications. Peptide identifications were accepted if they could be established at greater than 98.0% probability to achieve an FDR less than 1.0% by the Peptide Prophet algorithm (Keller and colleagues) with Scaffold delta-mass correction. Protein identifications were accepted if they could be established at greater than 99.0% probability to achieve an FDR less than 1.0% and contained at least 2 identified peptides. Protein probabilities were assigned by the Protein Prophet algorithm [68]. Proteins that contained similar peptides and could not be differentiated based on MS/MS analysis alone were grouped to satisfy the principles of parsimony.

## Treg suppression assay

Suppression assays were set up according to Collison & Vignali [69]. Briefly, Tregs from naïve DO-WT and DO-KO mice were isolated from the spleens of 4–6-week-old mice using a CD4$^+$CD25$^+$ isolation kit (**StemCell Technologies,** Vancouver, Canada; **Treg Isolation Kit**). Naïve CD4$^+$ responder T cells (Tconv) were magnetically isolated (**StemCell Technologies (**Vancouver, Canada)**, Naïve CD4 Isolation Kit**) from DO-WT mice and labeled with Cell-Trace Violet (**ThermoFisher Scientific,** Waltham MA) according to the manufacturer's conditions. CD28 stimulation was provided by isolated APCs via Thy 1.2 depletion (**StemCell Technologies (**Vancouver, Canada)**, Thy1.2 Isolation Kit**) before irradiation (3,000 rads). Soluble anti-CD3 (**Clone 2C11, eBiosciences (**San Diego, CA)**) was provided to all stimulation wells (1 ug/mL). The concentration of Tregs per well was varied so that the ratio of Treg:Tconv

cells per well were as follows: 1:1, 1:2, 1:4, and 1:8. Cells were cultured for 72 hours before FACS analysis. Percent suppression was calculated with the following formula:

$$\% \; Suppression = \frac{(\% \; of \; proliferated \; responders \; with \; no \; Treg - \% \; of \; proliferated \; responders)}{\% \; of \; proliferated \; responders \; with \; no \; Treg}$$

### Immunization and modified MLR

B6 or DO-KO mice (6–10 weeks old) were scarified and spleens harvested, processed, and resuspended in sterile PBS at $200 \times 10^6$ cell/mL. Recipient mice were then IP injected with $30 \times 10^6$ cells per mouse. On Day 8–9 postimmunization, splenocytes from immunized mice were harvested and used as responder cells for in vitro restimulation experiments. Stimulator B cells for the in vitro restimulation experiments were isolated from either DO-WT or DO-KO spleens and by negative selection (**StemCell Technologies, B cell isolation kit**) and treated with 200 ng/mL of LPS for 48 hours before irradiation with 800 rads to destroy any contaminant T cells. Responder and stimulator cells were cocultured at the ratio of 1:9 (0.5 million responder cells to 4.5 million stimulator B cells); culture time was dependent on the experiments.

### Cell proliferation assay

To measure CD4 T-cell responses in the modified MLR, cultured responder cells were harvested one week after the first round of in vitro restimulation and stained with Celltrace CFSE proliferation dye (**ThermoFisher Scientific,** Waltham, MA) before mixing with fresh stimulator B cells, prepared as above, for a second round of in vitro restimulation. On Day 5–6 after the second in vitro restimulation, cultured cells were harvested and stained with fluorescent antibodies for flow cytometry. To assess the percentage of precursor cells responding to either stimulator strain, proliferation modeling was performed using the Cell Tracking Wizard mode of ModFit LT (5.0). The PF represents the fraction of cells in each culture condition that experienced at least one round of division due to antigen stimulation.

### TCR sequencing

cDNA from cell lysates was subjected to TCR-beta CDR3 deep sequencing (Adaptive Technologies, Seattle, WA). As naïve TCR clonal controls, live CD3+ CD4+ T cells from spleens of 6–10-week-old naïve DO-WT or DO-KO mice were also sorted and deep sequenced.

### CD4 T-cell PF studies

Class II tetramer enrichment strategy was performed per Moon and colleagues [70]. Briefly, cells from all the peripheral lymphoid organs (spleens and all the lymph nodes) were harvested from 6–10-week-old naïve DO-WT or DO-KO and DR1+ DO-WT or DR1+ DO-KO mice. Cells were stained with PE-conjugated MOG tetramers or PE-conjugated CII tetramers for 2 hours at 37°C in RPMI + 2% FBS + 0.1% azide in the presence of 50 nM Dasatinib (**Cell Signaling Technology**, Danvers, MA). Tetramer positive cells were isolated via anti-PE bead positive selection (**Stemcell Technologies, PE positive isolation kit**) and subjected to flow cytometry analysis.

### Disease (EAE and CIA) induction

**EAE.** To induce EAE in B6 and DO-KO mice, 8–10-week-old mice received 2 subcutaneous injections of MOG$_{35-55}$ (150 ug/injection) in CFA (4 mg/mL *M.Tb*) at the base of the tail and between the shoulder blades, along with 1 injection of pertussis toxin (200 ng/mouse) IP

on Day −2. Two days later (Day 0), immunized mice received another 200 ng injection of pertussis toxin IP. Mice were visually scored by a blinded second party for the development of neurological symptoms starting from Day 0. Disease development was broken down as follows: 0 = no visible motor impairment, 1 = limp tail, 2 = limp tail + partial hind limb paralysis, 3 = limp tail + total hind limb paralysis, 4 = limp tail, total hind limb paralysis + partial fore limb paralysis.

**CIA.** To induce CIA in DR1[+] DO-WT or DR1[+] DO-KO mice, male mice (8–10 weeks old) received one intradermal (in tail) injection of 100 μg of bovine CII protein (2 mg/mL in 0.05 M acetic acid) in CFA (4 mg/mL of *M.Tb*). On Day 21 postimmunization, mice received a second intradermal tail immunization with collagen (2 mg/mL in 0.05 M acetic acid) in CFA (4 mg/mL *M.Tb*). Mice were visually monitored for the development of swollen joints.

## In vivo near-IR fluorescence imaging

**EAE imaging.** Both diseased and naïve control DO-WT and DO-KO mice were co-injected IV with dye-labeled anti-CD4 (CD4-IRDye800CW) and anti-MBP (MBP-IR-Dye68RD). Mice were killed at 72 hours post-injection and perfused with 10 mL cold PBS to remove blood from the CNS tissue. Internal organs were removed and skin trimmed away to allow for spinal column visualization. Skin around the inguinal and brachial lymph nodes was retained as a positive control for CD4 staining. Mice were then imaged using a Li-core Pearl imager (**LI-COR Biosciences**, Lincoln, NE). Co-localization was determined by the intensity of yellow signal on the merged image.

**CIA imaging.** Diseased CIA mice were injected with 4 nmol of Collagen probe (CMP-IR-Dye680RD) IV and anti-CD4 antibody (anti-CD4-IRDye800CW) IP. Forty-eight hours later, diseased mice were humanely killed and imaged ex vivo (skin off) using a Li-core Pearl imager (**LI-COR Biosciences**, Lincoln, NE). Co-localization was determined by the intensity of yellow signal on the merged images.

## Ex vivo recovery of CII tetramer–specific cells

Arthritic mice were injected with 4 nmol of Collagen probe (CMP-IRDye680RD) IV and 150 μg of IRDye800CW-conjugated CII tetramer intradermally into the footpad. Mice were imaged at 1 hour, 24 hours, 48 hours, and 72 hours post-injection to follow the CII tetramer distribution in vivo. At 72 hours post-injection, diseased mice were humanely killed and the popliteal lymph nodes harvested. Recovered lymphocytes from the popliteal lymph nodes were then stained with fluorescent antibodies and live/dead dye listed above for flow cytometry.

## Ex vivo recovery of tissue infiltrating cells from CNS of EAE-diseased mice

Diseased DO-WT and DO-KO mice were perfused through the left cardiac ventricle with 10 mL of cold PBS to remove blood from the brain and spinal cord tissue. After perfusion, brains were dissected and spinal cords flushed from the column using an 18G 1.5 needle. Tissue was digested with Collagenase D (2.5 mg/mL [Roche]) and DnaseI (1mg/mL [Sigma]) in DMEM for 45 minutes at 37°C. Mononuclear cells were isolated using a 70%/37% percoll gradient (500*g* for 20 minutes at room temperature, brake off).

## Detection of altered MOG$_{35-55}$ presentation using 2D2 Tg mice

EAE was induced as previously described in both DO-WT and DO-KO mice. Mice were humanely killed on Day 17 of disease, brains and spinal columns were pooled, and infiltrating

mononuclear cells were isolated as described above. Infiltrating APCs were then isolated by removal of CD90.2$^+$ cells (**StemCell Technologies**, Vancouver, Canada). Peripheral APCs from the draining lymph nodes of diseased DO-WT and DO-KO mice were also isolated via depletion of CD90.2$^+$ cells (**StemCell Technologies**, Vancouver, Canada). Both infiltrating APCs and peripheral APCs were irradiated (800 rads) to prevent proliferation during the culture time. Responder CD4 T cells were isolated (**StemCell Technologies**, Vancouver, Canada) from the spleens of female 2D2 transgenic mice and resuspended in complete RPMI media. A total of $1 \times 10^5$ responder 2D2 CD4 T cells were plated in triplicates with either (a) infiltrating APCs ($0.25 \times 10^5$, $0.5 \times 10^5$, and $1 \times 10^5$) or (b) draining LN B cells ($0.25 \times 10^5$, $0.5 \times 10^5$, and $1 \times 10^5$) in complete RPMI media. Cells were cultured for 48 hours at 37°C before surface staining and FCS analysis.

### MOG$_{35-55}$ concentration curve

CD4 T cells from 2D2 transgenic mice were isolated as described above and labeled with Cell-Trace Violet (**Molecular Probes**, Eugene, OR) per manufacturer's specifications before coculture with irradiated naïve splenic DO-WT or DO-KO B cells (800 rads). Isolated B and T cells were plated at a 1:1 ratio for a total of $2 \times 10^5$ cells per well. Various concentrations of MOG$_{35-55}$ (0, 0.01, 0.1, 0.5, 1, 10 μM) were then added for 48 hours at 37°C before surface staining and FACS analysis. All experimental conditions were set in triplicates.

### Statistical analysis

All statistical analysis was performed using Graphpad Prism (v8, San Diego, CA). Simpson's Diversity Index and Productive Rearrangement calculations were performed using Adaptive Biotechnologies (Seattle, WA) Tools. Unless otherwise noted, all statistical tests were standard Student $t$ tests. Error bars represent mean ± SEM.

## Supporting information

**S1 Fig. Number of peptides identified by mass spectrometry in two replicate experiments.** Lower numbers correspond to the first experiment utilizing 5 mice per group (944 KO versus 1,254 WT). Upper numbers correspond to a replicate experiment using 10 mice per group (1,290 KO versus 1,666 WT). Experimental results depicted in this figure can be found in S1 Data. KO, knockout; WT, wild-type.
(TIF)

**S2 Fig. DO-KO–derived Tregs have a similar suppressive capacity as DO-WT–derived Tregs.** Tregs from naïve DO-WT (white) and DO-KO (red) mice were plated at varying concentrations (1:1, 1:2, 1:4, 1:8) with proliferation dye–labeled naïve CD4 T cells in the presence of irradiated APCs with soluble CD3 for 72 hours. The amount of suppression was calculated as follows: $\% \, Suppression = \frac{(\% \, of \, proliferated \, responders \, with \, no \, Treg - \% \, of \, proliferated \, responders)}{\% \, of \, proliferated \, responders \, with \, no \, Treg} * 100$. Representative curve of 3 individual replicate suppression assays. Experimental results depicted in this figure can be found in S9 Data. APC, antigen-presenting cell; DO, H2-O; KO, knockout; Treg, regulatory T cell; WT, wild-type.
(TIF)

**S3 Fig. The DR1$^+$DO-KO lymphoid compartment is devoid of any gross alterations.** Similar cellular distribution in lymphoid tissues in naïve DR1$^+$DO-WT (Blue) and DR1$^+$DO-KO (Red) mice. Compiled graphs showing characterization data from 4 individual DR1$^+$DO-WT and 4 DR1$^+$DO-KO mice. Experimental results depicted in this figure can be found in S10

Data. DO, H2-O; DR1, HLA-DR1; KO, knockout; WT, wild-type.
(TIF)

**S4 Fig. Proposed differential impacts of DO on the outcome of CD4 T cells in thymic negative selection.** Illustration of proposed model of how negative selection would be impacted in the presence of DO (left) or in its absence (right). A higher density of the cognate epitope leads to successful negative selection and, conversely, absence of DO would cause a faulty negative selection. DO, H2-O.
(TIF)

**S5 Fig. EAE-diseased brains of DO-KO mice contained higher numbers of lymphocytes, but the total brain weights remained unchanged.** Total number of lymphocytes recovered from brains of diseased DO-KO or DO-WT mice. Diseased mice from all experiments (top left) or diseased mice from the 10–12-week time point (peak differences in disease scores, top right). Diseased brains of DO-KO and DO-WT mice weighed the same (bottom left). Each dot represents an individual mouse. Data represented as mean ± SEM. Experimental results depicted in this figure can be found in S11 Data. DO, H2-O; EAE, experimental autoimmune encephalomyelitis; KO, knockout; WT, wild-type.
(TIF)

**S6 Fig. Peptide pulsed B cells from either DO-WT or DO-KO mice induced similar levels of activation in 2D2 T cells.** Isolated B cells from DO-KO (Red) and DO-WT (White) were pulsed with various concentrations of $MOG_{35-55}$ peptide and cultured with isolated 2D2 CD4 T cells for 48 hours. Cells were then assessed for activation by up-regulation of CD69 in both groups. As shown, 2D2 T cells cocultured with B cells from either strain led to an almost linear increase in CD69 expression, indicating no differences in the level of I-A(b) between the two strains. Experimental results depicted in this figure can be found in S12 Data. DO, H2-O; KO, knockout; MOG, myelin oligodendrocyte glycoprotein; WT, wild-type.
(TIF)

**S1 Table. All shared peptides eluted from DO-WT and DO-KO mice.** Identified peptides were clustered based on the core sequence. For each core sequence, the number of times seen in the MassSpec was then combined to obtain a quantitative value. Each core sequence was then run through the IEDB I-A$^b$ binding prediction algorithm to gain an idea of relative binding strengths. DO, H2-O; IEDB, Immune Epitope Database; KO, knockout; MassSpec, mass spectrometry; WT, wild-type.
(XLSX)

**S2 Table. Unique peptides identified only in samples from DO-WT mice.** Identified peptides were clustered based on the core sequence. For each core sequence, the number of times seen in the MassSpec was then combined to obtain a quantitative value. Each core sequence was then run through the IEDB I-A$^b$ binding prediction algorithm to gain an idea of relative binding strengths. DO, H2-O; IEDB, Immune Epitope Database; MassSpec, mass spectrometry; WT, wild-type.
(XLSX)

**S3 Table. Unique peptides identified only in samples from DO-KO mice.** Identified peptides were clustered based on the core sequence. For each core sequence, the number of times seen in the MassSpec was then combined to obtain a quantitative value. Each core sequence was then run through the IEDB I-A$^b$ binding prediction algorithm to gain an idea of relative binding strengths. IEDB, Immune Epitope Database; KO, knockout; MassSpec, mass

spectrometry.
(XLSX)

**S1 Data. IEDB affinity prediction.** Peptide sequences identified in both DO-WT and DO-KO mice by MassSpec from 2 independent I-A$^b$ elution experiments were pooled for analysis. Because of the open-ended nature of the MHC class II molecule, peptide sequences were collapsed into just the core amino acids for each peptide. The core sequence was then run through the MHC-II binding predictions function on the IEDB. All core sequences can be found in S1–S3 Tables. Lower the affinity scores (i.e., closer to 0) indicate peptides with higher binding affinities to the I-A$^b$ molecule. These data are depicted in Fig 1B, S1 Fig, and S1–S3 Tables. DO, H2-O; IEDB, Immune Epitope Database; KO, knockout; MassSpec, mass spectrometry; WT, wild-type.
(XLSX)

**S2 Data. Naïve characterization of DO-WT and DO-KO mice spleen from 8 naïve DO-WT and DO-KO mice were harvested and total cells isolated.** Cell numbers were obtained via counting in hemocytometer chambers. Cells were then stained for the detection of CD4 T cells (Live/Dead dye$^-$B220$^-$CD11c$^-$F480$^-$CD3$^+$CD4$^+$CD8$^-$), CD8 T cells (Live/Dead dye$^-$B220$^-$CD11c$^-$F480$^-$CD3$^+$CD4$^-$CD8$^+$), CD4$^+$CD25$^+$ T cells (Live/Dead dye$^-$CD3$^+$B220$^-$CD11c$^-$F480$^-$CD8$^-$CD4$^+$CD25$^+$). In 3 separate experiments ($N$ = 21 mice total), DO-WT and DO-KO mice were also checked for irregularity in development of Tregs (Live/Dead dye$^-$CD3$^+$B220$^-$CD11c$^-$F480$^-$CD8$^-$CD4$^+$Foxp3$^+$). Total cell numbers for each of the individual populations were obtained by applying the FlowJo generated percentages to the total number of splenocytes. Statistical significance was calculated using GraphPad Prism, unpaired $t$ test, $P$ value $<$ 0.05. Error is represented as ±standard error mean (±SEM). These data are depicted in Fig 2. DO, H2-O; KO, knockout; Treg, regulatory T cell; WT, wild-type.
(XLSX)

**S3 Data. MLR and Naïve DO-WT and DO-KO TCR-B Sequencing Data.** (A) Individual replicates of the MLR experiment. CD4 T cells were identified as being: Live/Dead Dye- B220$^-$ CD19$^-$ F480$^-$ CD8$^-$ CD4$^+$. Proliferation was assessed by the percentage of CFSE dilution after coculture with B cells of the opposite strain. These data are in support of the representative plot in Fig 3A. (B) Individual replicates of the MLR experiment. CD4 T cells were identified as being Live/Dead Dye$^-$ B220$^-$ CD19$^-$ F480$^-$ CD8$^-$ CD4$^+$. Proliferation was assessed by the percentage of CFSE dilution after coculture with autologous B cells. These data are in support of the representative plot in Fig 3B. (C) Eight of the 12 individual MLR experiments shown in (A) were run through the Cell Tracking function of the ModFit LT software (Verity Software House). Percent PF (%PF) was predicted for CD4$^+$ T cells (Live/Dead Dye$^-$ B220$^-$ CD19$^-$ F480$^-$ CD8$^-$ CD4$^+$). Statistical significance was calculated using GraphPad Prism, unpaired $t$ test, $P$ value $<$ 0.05, ±SEM. These data are depicted in Fig 3C. (D) %PF was predicted using the Cell Tracking function of the ModFit LT software (Verity Software House) for CD4$^+$ T cells (Live/Dead Dye$^-$ B220$^-$ CD19$^-$ F480$^-$ CD8$^-$ CD4$^+$), which received autologous B cell stimulation. Statistical significance was calculated using GraphPad Prism, unpaired $t$ test, $P$ value $<$ 0.05, ±SEM. These data are depicted in Fig 3D. (E) TCR-B sequences from DO-WT and DO-KO mice were run through the Differential Abundance analysis tool available on the Adaptive Biotechnologies (Seattle, WA) website using the default settings: minimum # of template copies need to be considered for analysis = 10, $P$ value $<$ 0.01, and two-sided binomial analysis with the Benjamini-Hochberg correction applied. These data are depicted in Fig 3E. (F, G, and TCR-B Details) All identified TCR-B amino acid sequences used for the naïve DO-WT and DO-KO analysis are available in S1_Data: Naïve KO_WT TCR-B Details.

Productive rearrangements and Simpson's Diversity (1/D) were calculated using the Diversity metrics tool available on the Adaptive Biotechnologies (Seattle, WA) https://www.adaptivebiotech.com. Data are reported in Fig 3F and 3G. CFSE, Carboxyfluorescein succinimidyl ester; DO, H2-O; KO, knockout; MLR, mixed lymphocyte reaction; PF, precursor frequency; TCR-B, T-cell receptor beta chain; WT, wild-type.
(XLSX)

**S4 Data. Naïve PF of collagen (CII)–specific CD4 T cells in DR1+DO-WT and DR1+DO-KO mice.** (A) CII-specific CD4 (Live/Dead Dye− B220− CD11c− F480− CD8− CD4+CII Tetramer+) T cells were enriched from total naïve splenocytes via anti-PE bead pull-down after cells were labeled with CII(289–294)/DR1 tetramer. The total number of CII-specific CD4 T cells were calculated as described by Moon and colleagues [70]. Statistical significance was calculated using GraphPad Prism, unpaired $t$ test, $P$ value < 0.05, ±SEM. These data are depicted in Fig 4A. (B) Five naïve DR1+DO-WT and DR1+DO-KO mice were subcutaneously immunized with 100 μg of CII protein + CFA (1 mg/mL). Seven days postimmunization draining lymph nodes were harvested and pooled and stained for CII specificity: Live/Dead Dye− B220− CD11c− F480− CD8− CD4+CII Tetramer+. These data are depicted in Fig 4B. No statistical analysis was performed due to pooling of mice. CFA, Complete Freunds Adjuvant; CII, type II collagen; DO, H2-O; DR1, HLA-DR1; KO, knockout; PE, phycoerythrin; PF, precursor frequency; WT, wild-type.
(XLSX)

**S5 Data. In vivo labeling of CII-specific CD4 T cells from CIA diseased mice.** Draining lymph nodes from CIA diseased DR1+DO-WT and DR1+DO-KO mice were harvested and the total number of CII specific CD4 T cells (Live/Dead Dye− B220− CD11c− F480− CD8− CD4+CII Tetramer+) was assessed by flow cytometry. Total cell numbers were obtained by applying the CD4+CII+ percent to the total number of cells recovered from the draining lymph nodes. Statistical significance was calculated using GraphPad Prism, unpaired $t$ test, $P$ value < 0.05, ±SEM. These data are depicted in Fig 5C. CIA, collagen-induced arthritis; CII, type II collagen; DO, H2-O; DR1, HLA-DR1; KO, knockout; WT, wild-type.
(XLSX)

**S6 Data. Naïve PF of MOG-specific CD4 T cells in DO-WT and DO-KO mice.** (A) MOG-specific CD4 (Live/Dead Dye− B220− CD11c− F480− CD8− CD4+MOG Tetramer+) T cells were enriched from total naïve splenocytes via anti-PE bead pull-down after cells were labeled with MOG(35–55)/I-A$^b$ tetramer. The total number of MOG-specific CD4 T cells was calculated as described by Moon and colleagues [70]. Statistical significance was calculated using GraphPad Prism, unpaired $t$ test, $P$ value < 0.05, ±SEM. These data are depicted in Fig 6A. (B) Three naïve DO-WT and DO-KO mice were subcutaneously immunized with 100 μg of MOG (35–55) peptide + CFA (1 mg/mL $M.$ $tuberculosis$). Seven days postimmunization draining lymph nodes were harvested and pooled and stained for MOG specificity: Live/Dead Dye− B220− CD19− F480− CD8− CD4+MOG Tetramer+. Statistical significance was calculated using GraphPad Prism, unpaired $t$ test, $P$ value < 0.05, ±SEM. These data are depicted in Fig 6B. CFA, Complete Freuds Adjuvant; DO, H2-O; KO, knockout; MOG, myelin oligodendrocyte glycoprotein; PE, phycoerythrin; PF, precursor frequency; WT, wild-type.
(XLSX)

**S7 Data. EAE disease development in DO-WT and DO-KO mice.** (A) Visual scores of increasing paralyses in EAE-diseased DO-WT and DO-KO mice. Mice were immunized as described in Materials and methods and monitored daily for developing paralysis. Statistical significance was calculated using GraphPad Prism, two-way ANOVA, $P$ value < 0.05, ±SEM.

These data are depicted in Fig 7A. (C) Percentage of CNS infiltrating MOG-specific CD4+ (Live/Dead Dye− B220− CD11c− F480− CD8− CD4+MOG Tetramer+) cells recovered from individual mice after 10–12 days of EAE development. The total number of CNS infiltrating MOG-specific CD4+ T cells was generated by applying the percentage of MOG+CD4+ cells to the total number of CNS infiltrating lymphocytes. Statistical significance was calculated using GraphPad Prism, unpaired $t$ test, $P$ value < 0.05, ±SEM. These data are depicted in Fig 7C. (D) Percentage of CNS infiltrating MOG-specific CD4+ (Live/Dead Dye− B220− CD11c− F480− CD8− CD4+MOG Tetramer+) cells recovered from individual EAE-diseased DO-WT and DO-KO mice. The total number of CNS infiltrating MOG-specific CD4+ T cells was generated by applying the percentage of MOG+CD4+ cells to the total number of CNS infiltrating lymphocytes. Statistical significance was calculated using GraphPad Prism, unpaired $t$ test, $P$ value < 0.05, ±SEM. These data are depicted in Fig 7D. CNS, central nervous system; DO, H2-O; EAE, experimental autoimmune encephalomyelitis; KO, knockout; MOG, myelin oligodendrocyte glycoprotein; WT, wild-type.
(XLSX)

**S8 Data. Alterations in MOG(35–55) presentation by DO-KO CNS infiltrating APCs caused decreased CD69 expression by MOG-specific CD4 T cells.** APCs were isolated from either CNS tissue (B) or draining lymph nodes (C) of EAE-diseased DO-WT and DO-KO mice via Thy1.2 depletion (StemCell Technologies). MOG-specific CD4 T cells were isolated (CD4+ Isolation kit, StemCell Technologies) from the spleens of naïve 2D2 mice. Each condition was plated in triplicate. Cells were cocultured for 48 hours and CD69 expression was evaluated (Live/Dead Dye− B220− CD11c− F480− CD8− CD4+CD69+). Statistical significance was calculated using GraphPad Prism, unpaired $t$ test, $P$ value < 0.05, ±SEM. These data are depicted in Fig 8C and 8D. APC, antigen-presenting cell; CNS, central nervous system; DO, H2-O; KO, knockout; MOG, myelin oligodendrocyte glycoprotein.
(XLSX)

**S9 Data. DO-KO Tregs are not more suppressive than DO-WT Tregs.** Replicate data for the in vitro Treg suppression assay. CD4+CD25+ Tregs (StemCell Technologies) were isolated from DO-WT and DO-KO mice and cocultured with various concentrations of proliferation dye-stained conventional CD4 (CD4+CD25−) T cells (StemCell Technologies) isolated from a DO-WT mouse. Proliferation was assessed via proliferation dye dilution. Statistical significance was calculated using GraphPad Prism, unpaired $t$ test, $P$ value < 0.05, ±SEM. Data are depicted in S2 Fig. DO, H2-O; KO, knockout; Treg, regulatory T cell; WT, wild-type.
(XLSX)

**S10 Data. Naïve characterization of DR1+DO-WT and DR1+DO-KO mice.** Spleen from 4 naïve DO-WT and DO-KO mice were harvested and total cells isolated. Cells were stained for the detection of: CD3+ T cells (Live/Dead dye−B220−CD11c−F480−CD3+), double negative (DN) T cells (Live/Dead dye−B220−CD11c−F480−CD3+CD4−CD8−), CD4 T cells (Live/Dead dye−B220−CD11c−F480−CD3+CD4+CD8−), CD8 T cells (Live/Dead dye−B220−CD11c−F480−CD3+CD4−CD8+), and B cells (Live/Dead dye−B220+F480−CD11c−CD3−CD4−CD8−). Statistical significance was calculated using GraphPad Prism, unpaired $t$ test, $P$ value < 0.05. Error is represented as ±standard error mean (±SEM). These data are depicted in S3 Fig. DO, H2-O; DR1, HLA-DR1; KO, knockout; WT, wild-type.
(XLSX)

**S11 Data. EAE-diseased CNS tissue.** Individual data showing the total weight of 10 brains from diseased DO-WT and DO-KO mice. After PBS flushing, brains were weighed using a calibrated bench-top scale. The total number of CNS infiltrating lymphocytes was calculated by

applying the lymphocyte gate percentage generated in FlowJo to the total recovered mononuclear cell concentration calculated after Percoll gradient recovery. Statistical significance was calculated using GraphPad Prism, unpaired $t$ test, $P$ value $< 0.05$. Error is represented as ±standard error mean (±SEM). These data are depicted in S5 Fig. CNS, central nervous system; DO, H2-O; EAE, experimental autoimmune encephalomyelitis; KO, knockout; WT, wild-type.
(XLSX)

**S12 Data. CD69 up-regulation on 2D2 CD4 T cells after peptide pulsing DO-WT and DO-KO B cells.** Replicate data for MOG(35–55) peptide pulsing experiment indicating that both DO-WT and DO-KO B cells induce the same level of CD69 expression on MOG-specific CD4 T cells (Live/Dead Dye$^-$B220$^-$CD11c$^-$F480$^-$CD8$^-$CD4$^+$CD69$^+$) when pulsed with a known concentration of MOG(35–55) peptide. Statistical significance was calculated using GraphPad Prism, unpaired $t$ test, $P$ value $< 0.05$. Error is represented as ±standard error mean (±SEM). These data are depicted in S6 Fig. DO, H2-O; KO, knockout; MOG, myelin oligodendrocyte glycoprotein; WT, wild-type.
(XLSX)

## Acknowledgments

We thank Dr. Lars Karlsson and Dr. Peter Jensen for providing the DO-KO mice, Dr. Peter Calabresi for 2D2 Tg mice, Dr. Denis Zaller for the original HLA-DR1 mice, and Dr. Aeryon Kim for guidance in mice breeding and experimental design. In addition, we are grateful to Drs. Michael Yu and Lucas Bennink for the gift of the CMP-IRDye680RD; Drs. Nilabh Shastri, Martin Flajnik, Jay Bream, and Kevin Shenderov for helpful discussions and comments on the manuscript; the Bloomberg Flow Cytometry and Immunology Core Facility at the JHU School of Public Health for the use of their services; and the NIAID tetramer core facility for HLA-DR1 monomers and MOG (38–51)/I-A$^b$ tetramers.

## Author Contributions

**Conceptualization:** Robin A. Welsh, Nianbin Song, Scheherazade Sadegh-Nasseri.

**Data curation:** Robin A. Welsh, Nianbin Song.

**Formal analysis:** Robin A. Welsh, Nianbin Song, Catherine A. Foss, Tatiana Boronina.

**Funding acquisition:** Scheherazade Sadegh-Nasseri.

**Investigation:** Robin A. Welsh, Nianbin Song, Tatiana Boronina, Scheherazade Sadegh-Nasseri.

**Methodology:** Robin A. Welsh, Nianbin Song, Catherine A. Foss, Robert N. Cole, Scheherazade Sadegh-Nasseri.

**Resources:** Catherine A. Foss, Tatiana Boronina, Robert N. Cole.

**Supervision:** Scheherazade Sadegh-Nasseri.

**Validation:** Robin A. Welsh, Nianbin Song, Robert N. Cole.

**Visualization:** Catherine A. Foss, Scheherazade Sadegh-Nasseri.

**Writing – original draft:** Robin A. Welsh, Nianbin Song, Scheherazade Sadegh-Nasseri.

**Writing – review & editing:** Robin A. Welsh, Nianbin Song, Catherine A. Foss, Scheherazade Sadegh-Nasseri.

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
