## [Editor Report · Decision Letter 0]

1 Aug 2019

Dear Dr Sadegh-Nasseri, 

Thank you for submitting your manuscript entitled "Lack of the MHC Class II Chaperone H-2O Causes Susceptibility to Autoimmune Diseases" for consideration as a Research Article by PLOS Biology.

Your manuscript has now been evaluated by the PLOS Biology editorial staff as well as by an academic editor with relevant expertise and I am writing to let you know that we would like to send your submission out for external peer review.

*Please be aware that, due to the voluntary nature of our reviewers and academic editors, manuscripts may be subject to delays during the holiday season. Thank you for your patience.*

Please re-submit your manuscript within two working days, i.e. by Aug 03 2019 11:59PM.

Kind regards,

Di Jiang

PLOS Biology

---

## [Decision Letter · Decision Letter 1]

19 Sep 2019

Dear Dr Sadegh-Nasseri,

Thank you very much for submitting your manuscript "Lack of the MHC Class II Chaperone H-2O Causes Susceptibility to Autoimmune Diseases" for consideration as a Research Article at PLOS Biology. Your manuscript has been evaluated by the PLOS Biology editors, an Academic Editor with relevant expertise, and by three independent reviewers.

In light of the reviews (below), we will not be able to accept the current version of the manuscript, but we would welcome resubmission of a much-revised version that takes into account the reviewers' comments. Our Academic Editor advises you to fully address several unaccounted for results identified by reviewer 1 including the concern about the observation in Figure 2 that WT CD4 T cells mount a stronger proliferative response to KO splenocytes than the reverse, and she/he also suggests that you remove the results from 4 month cultured cells. We will not require that you test the peptide repertoire on DO KO medullary epithelium. We cannot make any decision about publication until we have seen the revised manuscript and your response to the reviewers' comments. Your revised manuscript is also likely to be sent for further evaluation by the reviewers.

Your revisions should address the specific points made by each reviewer. Please submit a file detailing your responses to the editorial requests and a point-by-point response to all of the reviewers' comments that indicates the changes you have made to the manuscript. In addition to a clean copy of the manuscript, please upload a 'track-changes' version of your manuscript that specifies the edits made. This should be uploaded as a "Related" file type. You should also cite any additional relevant literature that has been published since the original submission and mention any additional citations in your response. 

Before you revise your manuscript, please review the following PLOS policy and formatting requirements checklist PDF: http://journals.plos.org/plosbiology/s/file?id=9411/plos-biology-formatting-checklist.pdf. It is helpful if you format your revision according to our requirements - should your paper subsequently be accepted, this will save time at the acceptance stage.

Please note that as a condition of publication PLOS' data policy (http://journals.plos.org/plosbiology/s/data-availability) requires that you make available all data used to draw the conclusions arrived at in your manuscript. If you have not already done so, you must include any data used in your manuscript either in appropriate repositories, within the body of the manuscript, or as supporting information (N.B. this includes any numerical values that were used to generate graphs, histograms etc.). For an example see here: http://www.plosbiology.org/article/info%3Adoi%2F10.1371%2Fjournal.pbio.1001908#s5.

For manuscripts submitted on or after 1st July 2019, we require the original, uncropped and minimally adjusted images supporting all blot and gel results reported in an article's figures or Supporting Information files. We will require these files before a manuscript can be accepted so please prepare them now, if you have not already uploaded them. Please carefully read our guidelines for how to prepare and upload this data: https://journals.plos.org/plosbiology/s/figures#loc-blot-and-gel-reporting-requirements.

Upon resubmission, the editors will assess your revision and if the editors and Academic Editor feel that the revised manuscript remains appropriate for the journal, we will send the manuscript for re-review. We aim to consult the same Academic Editor and reviewers for revised manuscripts but may consult others if needed.

We expect to receive your revised manuscript within two months. Please email us (plosbiology@plos.org) to discuss this if you have any questions or concerns, or would like to request an extension. At this stage, your manuscript remains formally under active consideration at our journal; please notify us by email if you do not wish to submit a revision and instead wish to pursue publication elsewhere, so that we may end consideration of the manuscript at PLOS Biology.

When you are ready to submit a revised version of your manuscript, please go to https://www.editorialmanager.com/pbiology/ and log in as an Author. Click the link labelled 'Submissions Needing Revision' where you will find your submission record. 

Sincerely,

Di Jiang, PhD

Associate Editor

PLOS Biology

Reviewer remarks:

Reviewer #1: Welsh and associates propose that H2O, a molecule involved in peptide editing for MHC class II presentation, alters the peptide repertoire presented by class II molecules, thereby modulating CD4 T cell precursor frequencies and increasing susceptibility to autoimmune disease in two models. Unfortunately, this is a fairly confusing manuscript that contains a large number of unjustified data interpretations and unfounded conclusions, and that also fails to consider and acknowledge previous literature appropriately. Comments in detail:

• Although it is correct that in vivo studies of mice over-expressing or lacking H2O partly reveal complex effects (not surprisingly given that editing will modulate presentation of individual peptides in complex ways), the literature contains a fair amount of data strongly suggesting that H2O acts as competitor for H2M, thus inhibiting editing by H2M. A key structural paper in this context, published by Guce ad Stern, is not cited. 

• The notion that H2O modifies susceptibility to autoimmune disease has already been introduced by the demonstration that NOD mice overexpressing H2O are protected from type 1 diabetes, which would be expected to be mentioned.

• Peptide elution experiments are known to produce poorly reproducible results, so that multiple rather than a single experiment would be important to confirm the data in Fig. 1. Affinity data should be presented as mean IEDB scores rather than better or worse than CLIP. While it is technically difficult or impossible to elute peptides from TECs, the statement that use of splenic B cells in their place is justified by “similarity in antigen processing” would need to be supported by evidence or an appropriate citation.

• Fig. 2 shows that WT CD4 T cells mount a stronger proliferative response to KO splenocytes than the reverse. Given that WT splenocytes presented, according to Fig. 1, more unique peptides, they are expected to be more antigenic for KO CD4+ T cells rather than the reverse. Thus, this result is unexpected and should at least be discussed.

• While the experiment in Fig. 2 indeed shows that, not surprisingly, WT and KO present different peptide repertoires, this does not indicate that H2O-KO CD4+ repertoire includes autoreactive cells (it is also unclear why the same is not proposed for the reverse response). The unique peptides absent from KO splenocytes are more appropriately considered “allo-type” rather than “self” peptides. Thus, no conclusion regarding autoreactivity can be deduced from this experiment.

• The authors determined TCR repertoires after 4 months of continued restimulation without justifying this very long delay. Extensive culture and restimulation are likely to produce a TCR repertoire deviating largely from the in vivo repertoire, reducing the information that can be derived from these experiments.

• The authors interpret the somewhat higher number of unique TCRs found in H2O ko CD4+ T cells as evidence of lack of thymic deletion. However, since WT splenocytes express a much higher number of unique and therefore antigenic peptides, T cells from H2O ko mice are likely to engage a broader T cell repertoire in their response. Thus, at least two and probably additional explanations for this difference are possible. Moreover, no statistical evaluation is provided to show that this difference is significant. The additional interpretation of Fig. 2, proposing that TCR shared between restimulated and naïve T cells indicate self-reactivity, also lacks underpinning. The legend to Fig. 2 should be clarified.

• The presentation of data concerning responses to the collagen peptide is highly confusing. As cited by the authors, this peptide is DM-sensitive, i.e. its presentation is inhibited by DM. Since DO acts as inhibitor of DM according to solid in vitro data, its absence will enhance DM editing of the collagen peptide and thereby reduce its presentation. This is discussed in the context of thymic selection, considering the expression of DO limited to TECs and B cells. Surprisingly the authors then contend that the peptide will be presented in higher numbers in DO ko mice, without justifying this notion. The paper the reports slightly though not statistically significant higher numbers of T cell precursors recognizing the collagen peptide, as well as increased numbers after immunization and higher contacts of specific T cells with damaged collagen in vivo. To remain consistent with the (surprising) notion that the collagen peptide is presented in higher numbers in the ko mice, they then propose that higher presentation by peripheral APCs is responsible for greater T cell stimulation. As this is contrary to the lack of expression of DO in peripheral macrophages and dendritic cells, this interpretation would at least require a demonstration that DO-expressing cells are key to stimulating collagen-reactive T cells.

• Like the collagen model, the EAE model provides evidence suggesting that H2O is indeed modulating autoreactive responses, partly using interesting methods for monitoring of autoreactive T cells in vivo. However also in this model the interpretation is largely speculative and little convincing or even contrary to known facts about H2O expression and function. The authors expect the MOG peptide to be DM-resistant, however actually measuring its editing by DM would render data interpretation much more convincing. Globally the findings in the two models of autoimmune pathologies are consistent with the notion suggested by strong literature data (though still requiring experimental validation) that H2O limits editing of low affinity self-peptides by H2M in the thymus, thereby enhancing negative CD4+ T cell selection. The authors seem to prefer an alternative model in which H2O modulates antigen presentation in the periphery, however without providing any convincing evidence for such a model.

Reviewer #2: The report by Welsh et al analyses the role of HLA-DO or H2-DO in T cells selection and autoimmunity. HLA-DO or H2-DO is a non-classical MHC class II (MHCII) molecule that binds HLA-DM or H2-DM and blocks its catalytic activation and/or cooperates with DM for effective epitope selection . DM is also a non-classical MHCII molecule, which stabilizes and binds MHCII molecules to promote high affinity peptides loading for MHCII antigen presentation. In contrast to DM, DO expression is restricted to B cells and medullary epithelial cells (mTEC), antigen-presenting cells, which remove auto reactive T cells in the thymus. Thus, the absence of DO should result in a different outcome of negative selection and also impact the presentation of self-peptides by B cells and mTECs. This work is well written, interesting and new but relies on the use of B cells instead of mTECs. Page 10 the authors write: “Use of B cells instead of thymic medulla was justified for the expected similarity in antigen processing due to the expression of DO” but no reference is mentioned. Is antigen processing and presentation (antigen uptake, MHCII trafficking, proteases activity, DM, DO expression, Ii cleavage) really similar between these two APCs type? Could the authors provide some evidence for this, as it would greatly strengthen the paper? Would peptides eluted from DO-/-mTECs have also like DO-/- B cells worse affinity than that CLIP for IAb? What is the number of CD4+ and Treg cells in mice deficient for DO? In the CIA model, can the authors measure the CII-specific antibody response and score the disease? Will it be possible to test CII antigen presentation in DR1+DO-/-mice?

Reviewer #3: The study by Welsh et al addresses the role of the non-classical MHC class II molecule H2-O in susceptibility to the development of autoimmune disease. Using two different mouse models of autoimmunity, the authors show that the absence of H2-O increases disease susceptibility in two autoimmune disease models. The effects of H2-0 are attributed to, in a peptide-specific manner, differences in epitope presentation densities or in T cell selection. There are important nuances of H2-O function brought out by the study- peptide-specific effects on epitope presentation density or T cell selection. Overall, the studies shed light on the in vivo functions of H2-O and have implications for better understanding of known HLA-DO associations with human disease.

While the results on the mouse model of disease are quite clear-cut, the interpretations and language is confusing to the reader in some sections of the paper, which should be addressed as outlined below. There are some other experimental points that should also be addressed. Overall, an important study.

Specific points

Figure 1A: Did the individual replicates conform to the overall trend shown?

It would be useful to show the individual data as well as the combined data.

Figure 1B: predicted affinities are of limited value in the absence of experimental data. Figure 1B should be deleted and in the absence of direct experimental data, some statements of the discussion are not justified (for example, “Peptides eluted from DO-KO B cells, however, conformed to an overall poorer binding affinity than CLIP for I-Ab. This finding suggests that in the absence of DO, DM is not narrowing the peptide repertoire to only those with high binding affinities”). The focus should be on the overall similarities of the results of Figure 1A with data from reference 43.

Results of Figure 2B, left panel are very interesting, particularly compared to Figure 2B, right panel- the averaged data from multiple experiments should also be shown.

Figures 4A and 4B-representative data from how many experiments?

Figure 6D has a legend but not an associated Figure.

 “less than optimal editing by DM in the absence of DO” model (page 20 lines 315-316) should increase peptide diversity not reduce diversity. This sentence should be clarified or removed. 

Page 20, lines 316-317: Not clear how a lower diversity of eluted peptides from DO-KO is expected to lead to a three-fold higher level of DO-WT CD4 T cell proliferation. This sentence should also be clarified or deleted. Figure 2A types of experiments could be dominated by a few highly proliferative clones.

---

## [Editor Report · Decision Letter 2]

21 Nov 2019

Dear Dr Sadegh-Nasseri,

Thank you for submitting your revised Research Article entitled "Lack of the MHC Class II Chaperone H-2O Causes Susceptibility to Autoimmune Diseases" for publication in PLOS Biology. I have now obtained advice from the Academic Editor who has assessed your revision. 

Based on our Academic Editor's evaluation, we will probably accept this manuscript for publication, assuming that you will modify the manuscript to address two remaining points raised by the Academic Editor. First, please report the result in Figure 3 using programs that allow estimation of the % of precursor cells that divided instead of only quoting the % of recovered cells that have proliferated. Second, in addition to quoting the numbers of peptides that differ between the WT and DO KO B cells, please also report how many peptides differ between the each WT sample and each DO KO sample. Please also make sure to address the data and other policy-related requests noted at the end of this email.

We expect to receive your revised manuscript within two weeks. Your revisions should address the specific points made by each reviewer. In addition to the remaining revisions and before we will be able to formally accept your manuscript and consider it "in press", we also need to ensure that your article conforms to our guidelines. A member of our team will be in touch shortly with a set of requests. As we can't proceed until these requirements are met, your swift response will help prevent delays to publication.

Sincerely,

Di Jiang

PLOS Biology

ETHICS STATEMENT:

The Ethics Statements in the submission form and Methods section of your manuscript should match verbatim. Please ensure that any changes are made to both versions.

-- Please include the full name of the IACUC/ethics committee that reviewed and approved the animal care and use protocol/permit/project license. Please also include an approval number.

-- Please include the specific national or international regulations/guidelines to which your animal care and use protocol adhered. Please note that institutional or accreditation organization guidelines (such as AAALAC) do not meet this requirement.

DATA POLICY:

Regardless of the method selected, please ensure that you provide the individual numerical values that underlie the summary data displayed in the following figure panels as they are essential for readers to assess your analysis and to reproduce it: 1B, 2A-C, 3A-G, 4AB, 5C, 6AB, 7ACD, 8BC, S1, S2, S4, S5. NOTE: the numerical data provided should include all replicates AND the way in which the plotted mean and errors were derived (it should not present only the mean/average values).

---

## [Editor Report · Decision Letter 3]

13 Jan 2020

Dear Dr Sadegh-Nasseri,

On behalf of my colleagues and the Academic Editor, Dr. Philippa Marrack, I am pleased to inform you that we will be delighted to publish your Research Article in PLOS Biology. 

Early Version

PRESS 

Kind regards,

Krystal Farmer

PLOS Biology

on behalf of

Di Jiang,

Associate Editor

PLOS Biology